# Rebalancing Return Coverage for Conditional Sequence Modeling in Offline Reinforcement Learning

**Wensong Bai**[1], **Chufan Chen**[1], **Yichao Fu**[1], **Qihang Xu**[1], **Chao Zhang**[1,2,*] **Hui Qian**[1]

[1]College of Computer Science and Technology, Zhejiang University, Hangzhou, China
[2]Advanced Technology Institute, Zhejiang University, Hangzhou, China
{wensongb,chufan.chen,fuyichao,22351233,zczju,qianhui}@zju.edu.cn

## Abstract

Recent advancements in offline reinforcement learning (RL) have underscored the capabilities of conditional sequence modeling (CSM), a paradigm that models the action distribution conditioned on both historical trajectories and target returns associated with each state. However, due to the imbalanced return distribution caused by suboptimal datasets, CSM is grappling with a serious distributional shift problem when conditioning on high returns. While recent approaches attempt to empirically tackle this challenge through return rebalancing techniques such as weighted sampling and value-regularized supervision, the relationship between return rebalancing and the performance of CSM methods is not well understood. In this paper, we reveal that both expert-level and full-spectrum return-coverage critically influence the performance and sample efficiency of CSM policies. Building on this finding, we devise a simple yet effective return-coverage rebalancing mechanism that can be seamlessly integrated into common CSM frameworks, including the most widely used one, Decision Transformer (DT). The resulting CSM algorithm, referred to as Return-rebalanced Value-regularized Decision Transformer (RVDT), integrates both implicit and explicit return-coverage rebalancing mechanisms, and achieves state-of-the-art performance in the D4RL experiments.

## 1 Introduction

Offline RL aims to learn effective policies solely from previously collected datasets, without any further environment interaction [40, 17, 25, 30]. Recent advancements in offline RL primarily focus on the use of conditional sequence modeling (CSM), where the core idea is to learn the return-conditioned distribution of actions at each state given historical information [61, 25, 65, 66, 26]. One of the most representative methods in CSM is the decision transformer (DT) [11], wherein historical trajectories composed of multiple return-state-action triplets are exploited to train a Transformer-based model. During inference time, the model predicts a sequence of actions conditioned on a target return, effectively reframing the offline RL problem as a supervised learning task. By enabling the policy to make decisions directly from temporally extended trajectories, this paradigm circumvents the need for recursive value estimation, thereby avoiding both the challenges of Bellman completeness and the bootstrapping errors associated with dynamic programming-based methods [11, 68].

However, CSM-based algorithms encounter substantial difficulties when learning from suboptimal datasets, particularly due to the imbalance between near-optimal and low-return trajectories within datasets [34, 42]. This uneven distribution of trajectory quality in the training dataset may induce a

---

*Corresponding author: zczju@zju.edu.cn.

39th Conference on Neural Information Processing Systems (NeurIPS 2025).

significant distributional shift problem at inference time when conditioning on high returns, critically limiting the achievable performance of CSM methods [14, 10]. Recent advances have acknowledged such difficulties and proposed various mitigation strategies by rebalancing return distributions [48, 23, 42, 57, 25]. For instance, DiffStitch [42] employs diffusion models to synthesize high-return trajectories by stitching together optimal sub-segments drawn from multiple suboptimal trajectories. Critic-Guided Decision Transformer (CGDT) [57] and Q-value Regularized Transformer (QT) [25] integrate value-based guidance into the DT framework, implicitly assigning greater importance to near-optimal actions, thereby steering policies toward higher-return regions. These methods achieve enhanced performance through *explicit or implicit rebalancing of the return distribution*, and their efficacy has been empirically validated by achieving state-of-the-art (SOTA) results [42, 57, 25]. Nonetheless, none of these prior works provides a theoretical analysis elucidating the fundamental principles underpinning their empirical successes.

In this paper, we uncover the relationship between the return distribution of the training dataset and the performance of CSM policies at inference. Specifically, two aspects of return-coverage are relevant to the performance: (i) *expert-level return-coverage*, corresponding to the coverage of near-optimal and expert-level returns; (ii) *full-spectrum return-coverage*, which corresponds to the return-coverage over the entire return space[2]. By analyzing the coverage of these two returns within the offline datasets, we derive theoretical bounds quantifying the performance gap between the learned CSM policy (given fixed inference-time conditioning functions) and the optimal policy $\pi^*$ of the underlying MDP. Our analysis reveals that the performance gap is jointly influenced by both expert-level and full-spectrum return-coverage, providing a principled explanation for why return distribution rebalancing enhances policy performance and mitigates distributional shift at inference.

Building upon this finding, we propose a simple yet effective return-coverage rebalancing mechanism, which formulates rebalancing as a KL regularization between the agent's policy and an expert policy extracted from near-optimal trajectories in the dataset. The proposed mechanism is explicitly interpretable and seamlessly compatible with most existing CSM algorithms as a plug-in module. Leveraging this mechanism, we further design a practical CSM algorithm named Return-rebalanced Value-regularized Decision Transformer (RVDT), which achieves consistent performance improvements over state-of-the-art methods. Our contributions are summarized as follows:

1. We theoretically show how the performance of CSM-based policies is influenced by the coverage of expert-level and full-spectrum conditional returns in the offline training dataset. Specifically, we show that the gap between the expected return of the learned CSM policy and the optimal return is inversely proportional to the coverage of these two returns, and we derive an upper bound to explicitly characterize the associated sample complexity.

2. We propose an explicit return-coverage rebalancing mechanism as a plug-in module for enhancing existing CSM algorithms. In practice, an independent network is trained via imitation learning on expert-level trajectories to extract expert policies, whose KL divergence with the agent's policies is used as a regularizer. This regularizer has been shown to perform weighted sampling explicitly over trajectories when the policy is parameterized by a Gaussian distribution.

3. We design RVDT, which utilizes both explicit and implicit rebalancing mechanisms, effectively addressing the difficulties posed by limited expert data through the utilization of an expert KL divergence regularizer and a value-regularized guidance. This combination collectively mitigates inference-time distributional shift, enabling superior performance in challenging scenarios such as sparse-reward environments and low-quality datasets.

The performance of RVDT is evaluated on the D4RL benchmarks [16]. Experimental results demonstrate significant performance gains over representative baselines, establishing RVDT as a strong candidate for advancing the state-of-the-art (SOTA) in offline RL.

## 2 Related Work

Learning from imbalanced datasets, comprising predominantly low-return trajectories and relatively few high-return trajectories, has long been a critical challenge in offline RL [23, 21, 14, 10]. Broadly,

---

[2]The full-spectrum return-coverage is closely related to the runtime returns, i.e., the realistic returns during policy execution, as the actual distribution of runtime conditional returns during execution is typically uncertain.

prevailing methods tackling this challenge can be categorized into two types: *explicit rebalancing* and *implicit rebalancing*. Explicit methods encompass techniques such as dataset selection [12, 48, 32], weighted sampling [44, 21, 23], and data synthesis [62, 34, 42], while implicit methods generally leverage value estimation as additional guidance [38, 15, 17, 9, 57, 25].

**Sub-dataset Selection.** Several methods explicitly select informative subsets from imbalanced datasets, e.g., sample uncertainty [12, 48] or frequency of being forgotten [54] is utilized to prune the datasets. Other studies employ submodular functions that capture the notions of diversity, representation, and coverage to eliminate redundancy [52, 60, 59, 32]. A more straightforward approach, top-% BC, directly selects and fits only the highest-return trajectories within the dataset [39, 11].

**Weighted Sampling.** Another line of research adopts weighted sampling techniques to rebalance trajectory distributions [24, 41, 22, 44]. Hong et al. [21] leverage a Boltzmann distribution based on trajectory returns or advantages to reweight trajectories. Similarly, density-ratio weighting [23] utilizes importance sampling [49] to emulate sampling from other supplementary datasets.

**Data Augmentation and Generation.** Other methods address trajectory imbalance by augmenting or synthesizing data [56, 45, 64, 62, 34, 42]. For instance, QDT [62] relabels return tokens using a learned conservative Q-function, subsequently integrating these augmented trajectories into the training datasets. DiffStitch [42] leverages diffusion models to generate high-return trajectories by stitching optimal sub-segments extracted from multiple suboptimal trajectories.

**Value-regularized Supervised Learning.** Recent works such as BEAR [38], CDC [15], O-RL [7], TD3+BC [17], TD3+RKL [9], CGDT [57], and QT [25] integrate value-based guidance into supervised learning frameworks to prioritize high-return actions. As a result of leveraging transition-level value estimates to prioritize near-optimal actions without explicitly modifying the trajectory return distribution, these algorithms are considered as implicit return rebalancing methods.

## 3 Preliminary

### 3.1 Offline Reinforcement Learning

We model an agent-environment interaction as a finite horizon Markov Decision Process (MDP) $(\mathcal{S}, \mathcal{A}, \mathcal{R}, \mathcal{T}, \mu_0, H)$, where $\mathcal{S}$ and $\mathcal{A}$ denote the (continuous) state and action spaces. $\mathcal{R} : \mathcal{S} \times \mathcal{A} \to \mathbb{R}$ is the reward function, $\mathcal{T}$ defines the transition dynamics, i.e., $s_{t+1} \sim \mathcal{T}(\cdot|s_t, a_t)$. $\mu_0$ is the initial state distribution, i.e., $s_1 \sim \mu_0$, and $H$ is the time horizon. In the offline RL setup, algorithms are given a static dataset $\mathcal{D}$ consisting of trajectories $\tau = \{s_1, a_1, r_1, \cdots, s_H, a_H, r_H\}$, collected by a behavior policy $\pi_\beta$. Let $g(\tau_t) = \sum_{i=t}^{H} r_i$ (abbreviated as $g_t$) denote the return-to-go (RTG) of a trajectory at time step $t$, and $\mathcal{J}(\pi) = \mathbb{E}_{\tau \sim \pi}[g(\tau)]$ is the expected return of a policy $\pi$.

### 3.2 Return Conditioned Sequence Modeling

Motivated by the wide-ranging applications and remarkable success of transformer models in various domains [55, 8, 11, 27, 29, 36], CSM-based methods formulate policy learning as a return-conditioned supervised learning problem. Specifically, during training, the policy $\pi$ is optimized by minimizing the empirical negative log-likelihood (NLL) loss over a dataset $\mathcal{D}$ of trajectories:

$$\mathcal{L}(\pi) = -\sum_{\tau \in \mathcal{D}} \sum_{1 \le t \le H} \log \pi(a_t|s_t, g(\tau_t), \bar{\tau}_{t-1}^K), \tag{1}$$

where $\bar{\tau}_{t-1}^K = \{g_{t-K}, s_{t-K}, a_{t-K}, \ldots, g_{t-1}, s_{t-1}, a_{t-1}\}$ denotes the historical input with context length $K$. During inference, the learned policy $\pi$ is combined with a conditioning function $f(s)$ to form the inference-time policy $\pi_f(a|\mathbf{s})$ as $\pi_f(a|\mathbf{s}) := \pi(a|s, f(s), \bar{\tau}^K)$, where $\mathbf{s}$ denotes the collection of $s$ and historical triplets. Algorithms such as DT and its variants, e.g., QT [25] and CGDT [57], can be viewed as specific instantiations within this framework.

The NLL loss can be interpreted as approximating the conditional distribution $P_{\pi_\beta}(a|\mathbf{s}, g)$, where $P_{\pi_\beta}$ denotes the distribution over states, actions, and returns induced by the behavior policy $\pi_\beta$. In the regime of infinite data, it has been shown that the CSM objective effectively reweights the behavior policy based on the distribution of future returns. Under this setting, the optimal return-conditioned policy $\pi_f^{\text{CSM}}$ for a given conditioning function $f$ can be expressed as: $\pi_f^{\text{CSM}}(a|\mathbf{s}) =$

$\pi_\beta(a|\mathbf{s}) \frac{P_{\pi_\beta}(f(s)|\mathbf{s},a)}{P_{\pi_\beta}(f(s)|\mathbf{s})}$. With infinite data and a fully expressive policy class, optimizing (1) guarantees that the learned policy $\pi_f^{\mathrm{CSM}}$ is provably near-optimal [6].

**Theorem 1.** *Consider a finite-horizon MDP with horizon $H$, a behavior policy $\pi_\beta$, and a conditioning function $f$. Suppose the following conditions hold: (i) The return is sufficiently covered, i.e., $P_{\pi_\beta}(g = f(s_1)|s_1) \geq \alpha_f$ for all initial states $s_1$. (ii) The environment is nearly deterministic in the sense that for some functions $\mathcal{T}$ and $\mathcal{R}$, i.e., $P(r \neq \mathcal{R}(s,a)$ or $s' \neq \mathcal{T}(s,a)|s,a) \leq \epsilon, \forall s, a$. (iii) The conditioning function $f$ is consistent such that $f(s) = f(s') + r, \forall s$. Then the policy that optimizes the CSM objective (Eq. 1) is guaranteed to be near-optimal: $\mathbb{E}_{s_1} f(s_1) - J(\pi_f^{CSM}) \leq (1/\alpha_f + 2) H^2 \epsilon$.*

Theorem 1 establishes theoretical guarantees and success conditions for CSM methods. Brandfonbrener et al. [6] further analyze the performance gap between $\pi_{f^*}^{\mathrm{CSM}}$ and $\pi^*$, where $f^*$ corresponds to the optimal conditioning function. However, at inference time, the runtime conditional returns, computed as the prior returns minus sampled rewards $r$, cannot consistently remain optimal due to the inherent randomness of $r$ [5, 13]. This discrepancy between theory and practice substantially constrains its practical utility for enhancing the robustness and generalization of CSM algorithms.

## 4 Methodology

This paper focuses on investigating how return distribution of the training dataset impacts the performance of CSM policies through the lens of return-coverage. We theoretically reveal that the performance gap between the MDP's optimal policy $\pi^*$ and the learned CSM policy is jointly determined by both expert-level and full-spectrum return-coverage. Likewise, we establish that the sample complexity is similarly governed by these two aspects of coverage. The theoretical insights elucidate why previous approaches, utilizing explicit or implicit rebalancing of return distributions, effectively improve performance. Following this notion, we propose a simple yet effective return-coverage rebalancing mechanism, which is explicitly interpretable, and seamlessly compatible as a plug-in module with most existing CSM algorithms. To demonstrate the effectiveness of this mechanism, we integrate it into QT [25], the current SOTA, resulting in a new approach termed RVDT.

### 4.1 Analysis on Return-Coverage Rebalancing for Conditional Sequence Modeling

Due to the imbalance between expert-level and low-return trajectories in offline datasets, the study of Brandfonbrener et al. [6], which focuses on the optimal conditioning function $f^*$, becomes impractical for real-world scenarios. Our analysis extends this perspective by considering a more general function class that encompasses both potentially non-optimal runtime conditioning functions and optimal (or near-optimal) conditioning functions, denoted as $f$ and $f^*$, respectively. The runtime conditional return function $f$ corresponds to arbitrary possible returns that may be encountered during policy execution. Let $\mathcal{G}$ denote the collection of all possible returns collected by $\pi \in \Pi$, then $f : \mathcal{S} \to \mathcal{G}$. The target conditional function $f^*$ that we aim to find is the RTG under the optimal policy $\pi^*$, i.e., $f(s) = \max_\pi \mathbb{E}_\pi[g(s)]$. We define the expert-level return-coverage in the offline dataset generated by behavior policy $\pi_\beta$ as $P_{\pi_\beta}(g = f^*(s_1)|s_1)$, where the $f^*$ corresponds to the optimal policy $\pi^*$ of the underlying MDP. Correspondingly, the full-spectrum return-coverage is defined as $P_{\pi_\beta}(g = f(s_1)|s_1)$. We first analyze the performance of CSM policies under an idealized setting with infinite data and a fully expressive policy class, establishing an upper bound on the suboptimality of CSM policies and quantifying its dependence on both aspects of return-coverage.

**Theorem 2** (Performance gap with respect to return-coverage). *Consider a finite-horizon MDP with horizon $H$, behavior policy $\pi_\beta$, a runtime conditioning function $f$, and the optimal conditioning function for $\pi^*$ is $f^*$. Assume the following assumptions hold: (i) **Return-coverage:** $P_{\pi_\beta}(g = f(s_1)|s_1) \geq \alpha_f$ and $P_{\pi_\beta}(g = f^*(s_1)|s_1) \geq \alpha_f^*$ for all initial states $s_1$. (ii) **Near determinism:** $P(r \neq \mathcal{R}(s,a)$ or $s' \neq \mathcal{T}(s,a)|s,a) \leq \epsilon$ at all $(s,a)$ for some $\mathcal{T}$ and $\mathcal{R}$. (iii) **Consistency of $f$:** $f(s) = f(s') + r$ for all $s$. Then the following upper bound holds:*

$$J(\pi^*) - J(\pi_f^{CSM}) \leq (\frac{1}{\alpha_f^*} + 3)H^2\epsilon + (\frac{1}{\alpha_f} + \frac{1}{\alpha_f^*})H^2 C, \qquad (2)$$

*where $C \in (0, 1)$ is a constant. The proof is provided in Appendix A.2.*

The result formally characterizes how the effectiveness of policies trained via CSM is constrained by both expert-level and full-spectrum return-coverage. This implies that algorithms are capable of modulating $\alpha_f^*$ and $\alpha_f$ through rebalancing the return distribution, thereby facilitating the convergence of $\pi_f^{\text{CSM}}$ toward the optimal policy $\pi^*$. Notably, in the offline RL setting, the relationship $\alpha_f + \alpha_f^* \leq 1$ holds when $f \neq f^*$. In other words, strategies such as sub-dataset selection or weighted sampling that attempt to increase one type of return-coverage without altering the distribution's support tend to reduce the other.

The subsequent analysis examines the sample efficiency of CSM methods under common assumptions as in [6], with finite data and a restricted policy class. Letting $N$ denote the size of the dataset, Theorem 3 demonstrates that the sample complexity exhibits a structural dependence on the return-coverage analogous to that in Theorem 2. Note that for clarity, we consider the case of a single $s$ ($\bar{\tau}_{t-1}^K = \varnothing$) here, which can be readily extended to the non-empty $\bar{\tau}_{t-1}^K$ scenario.

**Theorem 3** (Sample complexity). *To get finite data guarantees, add to the above assumptions in Theorem 2 that (i) bounded occupancy mismatch: $P_{\pi_f^{\text{CSM}}}(s) \leq C_f \cdot P_{\pi_\beta}(s)$ for all $s$; (ii) the policy class $\Pi$ is finite; (iii) $|\log \pi(a|s,g) - \log \pi(a'|s',g')| \leq c$ for any $(a, s, g, a', s', g')$ and all $\pi \in \Pi$; (iv) the approximation error of $\Pi$ is bounded by $\epsilon_{approx}$, i.e., $\min_{\pi \in \Pi} L(\pi) \leq \epsilon_{approx}$.*

*Define the expected loss as $L(\hat{\pi}) = \mathbb{E}_{s \sim P_{\pi_\beta}} \mathbb{E}_{g \sim P_{\pi_\beta}(\cdot|s)} \left[ \text{KL} \left( P_{\pi_\beta}(\cdot|s,g) \,\|\, \hat{\pi}(\cdot|s,g) \right) \right]$. Then for any estimated CSM policy $\hat{\pi}_f$ that conditions on $f$ at inference time, with probability at least $1 - \delta$,*

$$J(\pi^*) - J(\hat{\pi}_f) \leq O\left( \left[ \frac{C_f}{\alpha_f} \sqrt{c} \left( \frac{\log |\Pi|/\delta}{N} \right)^{1/4} + \frac{C_f}{\alpha_f} \sqrt{\epsilon_{approx}} + \frac{\epsilon + C}{\alpha_f^*} + \frac{C}{\alpha_f} \right] H^2 \right). \tag{3}$$

*The proof, provided in Appendix A.3, utilizes uniform convergence results from supervised learning theory [50].*

Theorem 3 quantifies the sample complexity of CSM methods in terms of expert-level and full-spectrum return-coverage. Together with Theorem 2, our analysis unveils the fundamental principles underpinning the empirical successes of current return-coverage rebalancing strategies. That is, rebalancing $\alpha_f^*$ and $\alpha_f$ reduces the gap between the learned CSM policies and $\pi^*$.

## 4.2 Plug-in Return-Coverage Rebalancing Mechanism for CSM

Building upon the theoretical insights established in preceding theorems, we propose an explicit rebalancing mechanism based on KL divergence regularization with respect to a learned expert policy $\pi^e$. Specifically, a subset of near-optimal trajectories is filtered from the offline dataset, denoted as $\mathcal{D}_e \subset \mathcal{D}$, from which $\pi^e$ is extracted via imitation learning. Given a CSM policy $\pi_\theta$ parameterized by DT, our proposed rebalancing mechanism operates as a plug-in module appended to the original NLL loss in Eq. 1. We refer to the resulting CSM algorithm as Return-rebalanced Decision Transformer (RDT), which is optimized with the following loss:

$$\mathcal{L}_{\text{RDT}}(\theta) = \mathbb{E}_{\tau \sim \mathcal{D}} \Big[ \sum_{i=1}^{H} -\log \pi_\theta(a_i|s_i, g(\tau_i), \bar{\tau}_{t-1}^K) \Big] + \alpha \mathbb{E}_{\tau \sim \mathcal{D}_e} \sum_{i=1}^{H} \text{KL} \big[ \pi_\theta(\cdot|s_i, g(\tau_i), \bar{\tau}_{t-1}^K) \| \pi^e(\cdot|s_i) \big]. \tag{4}$$

The first term in Eq. (4) aligns the model with the empirical action distribution of $\pi_\beta$, while the regularization term guides the policy towards expert-level actions. Notably, the rebalancing module adopts KL divergence instead of MSE, designed to maintain compatibility with broader policy classes in modern CSM methods, such as diffusion models [69, 47].

To illustrate how the expert KL regularizer explicitly rebalances the return-coverage, we demonstrate that under a common stochastic policy parameterization, the regularization process can be equivalently interpreted as trajectory-level reweighting of expert-level samples in the offline dataset.

**Proposition 1** (KL regularization as weighted sampling strategy). *Assume the policy $\pi_\theta$ is parameterized by a factorized Gaussian distribution with a fixed standard deviation. Then optimizing the RDT objective in (4) is equivalent to optimizing the following weighted NLL loss (proof in A.4):*

$$\arg\min_{\pi_\theta} \mathcal{L}_{RDT}(\theta) = \arg\min_{\pi_\theta} \mathbb{E}_{\tau \sim \mathcal{D}} \Big[ (1 + \alpha \cdot \mathbb{I}[\tau \in \mathcal{D}_e]) \cdot \big( \sum_{i=1}^{H} -\log \pi_\theta(a_i|s_i, g(\tau_i), \bar{\tau}_{t-1}^K) \big) \Big]. \tag{5}$$

Proposition 1 shows that the KL regularization effectively performs a weighted resampling of expert trajectories, practically increasing the expert-level return-coverage within the offline dataset. Though an optimal rebalancing ratio exists, it is generally infeasible to compute explicitly. In practice, our rebalancing mechanism, akin to other rebalancing strategies [17, 9, 21], amplifies the gradient contributions of expert-level behaviors during training.

### 4.3 Return-rebalanced Value-regularized Decision Transformer

We integrate our proposed rebalancing mechanism into the current SOTA method, QT, resulting in a novel algorithm that incorporates both explicit and implicit return-coverage rebalancing mechanisms, referred to as RVDT. By implicitly rebalancing the return-coverage with a Q-value regularizer to DT, the loss function of QT is defined as follows.

$$\mathcal{L}_{\text{QT}}(\theta) = \mathbb{E}_{\tau \sim \mathcal{D}} \big[ \sum_{i=1}^{H} - \log \pi_\theta(a_i | s_i, g(\tau_i), \bar{\tau}_{t-1}^K) \big] - \eta \mathbb{E}_{\tau \sim \mathcal{D}} \mathbb{E}_{s_i \sim \tau, a_i \sim \pi_\theta} [Q^{\pi_\theta}(s_i, a_i)]. \quad (6)$$

However, accurate Q-value estimation can be prohibitive in complex environments, limiting the effectiveness of QT's value-based rebalancing mechanism. In contrast, RVDT uniquely combines explicit and implicit rebalancing mechanisms, i.e., an expert KL divergence regularizer and an value-regularized guidance, enhancing its adaptability to challenging offline datasets. The resulting learning objective for RVDT augments policy regularization with value-based guidance:

$$\mathcal{L}_{\text{RVDT}}(\theta) = \mathbb{E}_{\tau \sim \mathcal{D}} \big[ \sum_{i=1}^{H} - \log \pi_\theta(a_i | s_i, g(\tau_i), \bar{\tau}_{t-1}^K) \big] - \eta \mathbb{E}_{\tau \sim \mathcal{D}} \mathbb{E}_{s_i \sim \tau, a_i \sim \pi_\theta} [Q^{\pi_\theta}(s_i, a_i)] \\ + \alpha \mathbb{E}_{\tau \sim \mathcal{D}_e} \sum_{i=1}^{H} \text{KL} \big[ \pi_\theta(\cdot | s_i, g(\tau_i), \bar{\tau}_{t-1}^K) \| \pi^e(\cdot | s_i) \big]. \quad (7)$$

We adopt the same parameterization strategy as QT and employ offline DP to iteratively learn the value function $Q^{\pi_\theta}$. The coefficient $\eta$ follows a commonly adopted normalization scheme [17], i.e., $\eta = \frac{\bar{\eta}}{\mathbb{E}_{\tau \sim \mathcal{D}} \mathbb{E}_{(s_i, a_i) \sim \tau} |Q_\psi(s_i, a_i)|}$, where $\bar{\eta}$ is a tunable hyperparameter. The training and inference details, along with the algorithm pseudocode, are provided in Appendix B.

Compared to QT, RVDT not only inherits its strengths but also benefits from directly aligning its policy distribution with expert-level trajectories, thereby significantly enhancing performance in sparse-reward tasks where accurate Q-value estimation is infeasible. While empirical evidence (Section 5) confirms the superiority of RVDT, we provide a theoretical perspective to substantiate its superiority over related methods under idealized conditions (i.e., infinite data and neglecting approximation errors). The formal statement and proof of Proposition 2 are provided in Appendix A.5.

**Proposition 2** (Informal). *Let $\pi_e^*$, $\pi_w^*$, and $\pi_{qt}^*$ denote the optimal policies learned from the objectives of RVDT, RDT, and QT, respectively. Given the behavior policy $\pi_\beta$, there exists an expert-level subset $\mathcal{D}_e \subseteq \mathcal{D}$ and an appropriate coefficient $\alpha \geq 0$ such that the following inequalities hold for all states $s \sim d_{\pi_\beta}$: $\mathbb{E}_s V^{\pi_e^*}(s) \geq \mathbb{E}_s V^{\pi_w^*}(s) \geq \mathbb{E}_s V^{\pi_\beta}(s)$ and $\mathbb{E}_s V^{\pi_e^*}(s) \geq \mathbb{E}_s V^{\pi_{qt}^*}(s) \geq \mathbb{E}_s V^{\pi_\beta}(s)$.*

## 5 Experiment

We conduct a comprehensive empirical evaluation of our proposed approach on the D4RL benchmarks [16]. RVDT is compared against a broad range of baselines, mainly composed of *DP-based methods* [37, 19, 40], *value-regularized SL* [38, 7, 17], and *CSM approaches* [11, 51, 62, 28, 57, 63, 25]. To further assess the robustness of RVDT, we design a set of more challenging datasets to compare RVDT with the current SOTA method (QT) [25], focusing on three key desired properties: (i) effective utilization of limited expert data, (ii) learning ability under low-data regimes [46], and (iii) trajectory stitching ability. Finally, we perform carefully designed ablation studies to isolate and examine the contributions of key components in RVDT, including the KL regularization-based return-coverage rebalancing mechanism, the Q-value guidance, and the stochastic policy modeling.

**Benchmarks and Baselines.** We evaluate our method across five diverse domains from the D4RL benchmarks [16]: Gym, Adroit, Kitchen, Maze2D, and AntMaze, encompassing tasks from continuous control to multi-task manipulation and long-horizon planning. Our approach is compared against a broad set of baselines, including CSM-based methods (DT [11], StAR [51], QDT [62], GDT [28], CGDT [57], QT [25]), DP-based methods (IQL [37], BCQ [19], CQL [40]), value-regularized SL methods (BEAR [38], O-RL [7], TD3+BC [17]), and diffusion-based methods (Diffuser [31], Decision Diffuser [2], Diffusion-QL [58]). We further include a model-based approach (MoRel [33]) for broader comparison. QT is reproduced using its official implementation, while other baselines follow their original publications or results reported in QT's paper, depending on availability. Additional benchmark and implementation details are provided in Appendix C.1 and D.

### 5.1 Main Results on D4RL Benchmarks

The experimental results comparing RVDT against baselines on the D4RL benchmarks are presented in Table 1. RVDT consistently outperforms most baselines across various domains, establishing new SOTA performance on the majority of D4RL tasks. Task-level comparisons are provided below.

Table 1: Performance comparison across D4RL tasks. For each task, RVDT reports the mean and standard error of normalized scores [16] over 30 random rollouts and averaged over 5 random seeds.

| Gym Tasks | CQL | IQL | BCQ | TD3+BC | MoRel | BC | DD | DT | StAR | GDT | CGDT | QT | RVDT |
|---|---|---|---|---|---|---|---|---|---|---|---|---|---|
| halfcheetah-m-e | 91.6 | 86.7 | 69.6 | 90.7 | 53.3 | 55.2 | 90.6 | 86.8 | 93.7 | 93.2 | 93.6 | 93.2 | **94.4 ± 0.1** |
| hopper-m-e | 105.4 | 91.5 | 109.1 | 98.0 | 108.7 | 52.5 | 111.8 | 107.6 | 111.1 | 111.1 | 107.6 | 113.0 | **113.1 ± 0.5** |
| walker2d-m-e | 108.8 | 109.6 | 67.3 | 110.1 | 95.6 | 107.5 | 108.8 | 108.1 | 109.0 | 107.7 | 109.3 | 112.0 | **112.7 ± 1.6** |
| halfcheetah-m | 49.2 | 47.4 | 41.5 | 48.4 | 42.1 | 42.6 | 49.1 | 42.6 | 42.9 | 42.9 | 43.0 | 51.0 | **51.9 ± 0.3** |
| hopper-m | 69.4 | 66.3 | 65.1 | 59.3 | 95.4 | 52.9 | 79.3 | 67.6 | 59.5 | 77.1 | 96.9 | 99.6 | **100.2 ± 0.1** |
| walker2d-m | 83.0 | 78.3 | 52.0 | 83.7 | 77.8 | 75.3 | 82.5 | 74.0 | 73.8 | 76.5 | 79.1 | 87.2 | **90.2 ± 0.1** |
| halfcheetah-m-r | 45.5 | 44.2 | 34.8 | 44.6 | 40.2 | 36.6 | 39.3 | 36.6 | 36.8 | 40.5 | 40.4 | 48.8 | **53.8 ± 2.0** |
| hopper-m-r | 95.0 | 94.7 | 31.1 | 60.9 | 93.6 | 18.1 | 100.0 | 82.7 | 29.2 | 85.3 | 93.4 | 102.1 | **103.2 ± 1.9** |
| walker2d-m-r | 77.2 | 73.9 | 13.7 | 81.8 | 49.8 | 32.3 | 75.0 | 79.4 | 39.8 | 77.5 | 78.1 | 97.8 | **99.3 ± 0.8** |
| Average | 80.6 | 77.0 | 53.8 | 75.3 | 72.9 | 52.6 | 81.8 | 76.2 | 66.2 | 79.1 | 82.4 | 89.4 | **91.2** |

| Adroit Tasks | CQL | IQL | BCQ | BEAR | O-RL | BC | DD | D-QL | DT | StAR | GDT | QT | RVDT |
|---|---|---|---|---|---|---|---|---|---|---|---|---|---|
| pen-human | 37.5 | 71.5 | 66.9 | -1.0 | 90.7 | 63.9 | 66.7 | 72.8 | 79.5 | 77.9 | 92.5 | 111.9 | **127.2 ± 5.5** |
| hammer-human | 4.4 | 1.4 | 0.9 | 0.3 | 0.2 | 1.2 | 1.9 | 0.2 | 3.7 | 3.7 | 5.5 | 10.4 | **24.0 ± 1.5** |
| pen-cloned | 39.2 | 37.3 | 50.9 | 26.5 | 60.0 | 37.0 | 42.8 | 57.3 | 75.8 | 33.1 | 86.2 | 85.8 | **117.8 ± 8.6** |
| hammer-cloned | 2.1 | 2.1 | 0.4 | 0.3 | 2.0 | 0.6 | 1.7 | 3.1 | 3.0 | 0.3 | 8.9 | 11.8 | **21.3 ± 2.7** |
| Average | 20.8 | 28.1 | 29.8 | 6.5 | 38.2 | 25.7 | 28.3 | 33.4 | 40.5 | 28.8 | 48.3 | 55.0 | **72.6** |

| Kitchen Tasks | CQL | IQL | BCQ | BEAR | O-RL | BC | DD | D-QL | DT | StAR | GDT | QT | RVDT |
|---|---|---|---|---|---|---|---|---|---|---|---|---|---|
| kitchen-Comp. | 43.8 | 62.5 | 8.1 | 0.0 | 2.0 | 65.0 | 65.0 | 84.0 | 50.8 | 40.8 | 43.8 | 81.7 | **84.5 ± 2.3** |
| kitchen-partial | 49.8 | 46.3 | 18.9 | 13.1 | 35.5 | 33.8 | 57.0 | 60.5 | 57.9 | 12.3 | 73.3 | 72.5 | **75.0 ± 2.5** |
| Average | 46.8 | 54.4 | 13.5 | 6.6 | 18.8 | 49.4 | 61.0 | 72.2 | 54.4 | 26.6 | 58.6 | 77.1 | **79.8** |

| Maze2D Tasks | CQL | IQL | BCQ | BEAR | TD3+BC | BC | Diffuser | DD | DT | GDT | QDT | QT | RVDT |
|---|---|---|---|---|---|---|---|---|---|---|---|---|---|
| maze2d-u | 94.7 | 42.1 | 49.1 | 65.7 | 14.8 | 88.9 | 113.9 | 116.2 | 31.0 | 50.4 | 57.3 | 99.2 | **145.1 ± 3.8** |
| maze2d-m | 41.8 | 34.9 | 17.1 | 25.0 | 62.1 | 38.3 | 121.5 | 122.3 | 8.2 | 7.8 | 13.3 | 168.8 | **183.5 ± 4.5** |
| maze2d-l | 49.6 | 61.7 | 30.8 | 81.0 | 88.6 | 1.5 | 123.0 | 125.9 | 2.3 | 0.7 | 31.0 | 242.7 | **254.3 ± 4.6** |
| Average | 62.0 | 46.2 | 32.3 | 57.2 | 55.2 | 42.9 | 119.5 | 121.5 | 13.8 | 19.6 | 33.9 | 170.2 | **194.3** |

| AntMaze Tasks | CQL | IQL | BCQ | BEAR | TD3+BC | BC | DD | D-QL | DT | StAR | GDT | QT | RVDT |
|---|---|---|---|---|---|---|---|---|---|---|---|---|---|
| antmaze-u | 74.0 | 87.5 | 78.9 | 73.0 | 78.6 | 54.6 | 73.1 | 93.4 | 59.2 | 51.3 | 76.0 | 96.0 | **98.0 ± 4.0** |
| antmaze-u-d | 84.0 | 62.2 | 55.0 | 61.0 | 71.4 | 45.6 | 49.2 | 66.2 | 53.0 | 45.6 | 69.0 | 92.0 | **98.0 ± 4.0** |
| antmaze-m-d | 53.7 | 70.0 | 0.0 | 8.0 | 3.0 | 0.0 | 24.6 | **78.6** | 0.0 | 0.0 | 6.0 | 24.0 | 30.0 ± 6.3 |
| antmaze-l-d | 14.9 | 47.5 | 2.2 | 0.0 | 0.0 | 0.0 | 7.5 | **56.6** | 0.0 | 0.0 | 0.0 | 10.0 | 10.0 ± 0.0 |
| Average | 56.6 | 66.8 | 34.0 | 35.5 | 38.2 | 25.0 | 38.6 | **73.7** | 28.0 | 24.2 | 37.8 | 57 | 59.0 |

**Gym MuJoCo tasks** are widely employed to evaluate continuous control capabilities under dense reward conditions. From Table 1, we observe that RVDT achieves new SOTA results across all datasets. Notably, scores exceeding 100 on D4RL MuJoCo tasks are already near the theoretical performance ceiling, where boundary effects make even small improvements significant. Moreover, algorithms employing return-coverage rebalancing mechanisms, such as RVDT, QT, CGDT, and TD3+BC, consistently demonstrate superior performance over traditional methods (e.g., BC and DT) across the majority of tasks, providing empirical support for our theoretical analysis.

**Adroit tasks** present substantial challenges for offline RL due to pronounced extrapolation errors stemming from the limited coverage of human demonstrations [16]. Consequently, these tasks critically test robustness against distributional shift and the ability to imitate from sparse expert data. Several observations emerge from our results: (i) except for RVDT and QT, other baselines fail to achieve satisfactory performance; (ii) RVDT and QT significantly outperform DT due to their utilization of rebalancing mechanisms; (iii) RVDT notably surpasses QT with 25% and 105% improvements on pen and hammer, respectively. These findings underscore RVDT's effectiveness in learning from limited data and highlight the advantage gained from integrating explicit and implicit rebalancing mechanisms, effectively mitigating extrapolation errors and distributional shift problem.

**Kitchen tasks** primarily assess algorithms' capabilities in multi-task manipulation and generalization to unseen states. Our results indicate that RVDT slightly outperforms current SOTA methods, demonstrating its competence in managing multi-task dependencies and generalizing effectively. Furthermore, RVDT substantially surpasses non-CSM approaches, reinforcing the superiority of CSM methods over DP-based methods, aligning with findings from prior works [11, 25].

**Maze tasks** evaluate the trajectory stitching capacity of CSM algorithms [16, 25]. By integrating expert KL regularization and Q-value guidance into a Transformer-based policy, RVDT substantially

Table 2: Performance comparison under varying proportions of expert trajectory augmentation. Each setting (e.g., 15% RVDT) corresponds to training on a mixed dataset composed of the base dataset (merged `-m` and `-m-r`) and top-$k\%$ expert trajectories sampled from the `-m-e` dataset.

| Task | 0% Expert | | | 15% Expert | | | 30% Expert | | | 50% Expert | | |
|------|-----|-----|------|-----|------|------|------|------|------|------|------|------|
| | BC | QT | RVDT | BC | QT | RVDT | BC | QT | RVDT | BC | QT | RVDT |
| halfcheetah | 42.4 | 46.9 | **49.3** | 46.0 | 79.5 | **92.7** | 73.3 | 91.3 | **93.3** | 71.1 | 92.8 | **94.4** |
| hopper | 65.4 | 95.8 | **100.2** | 99.0 | 108.0 | **113.1** | 112.3 | 110.6 | **113.8** | 112.8 | 108.3 | **113.9** |
| walker2d | 81.0 | 92.3 | **92.7** | 107.8 | 106.3 | **110.3** | 108.2 | 108.0 | **112.4** | 108.1 | 113.6 | **113.8** |
| **Average** | 62.9 | 78.3 | **80.3** | 84.3 | 97.9 | **105.4** | 97.9 | 103.4 | **106.5** | 97.3 | 104.9 | **107.0** |

Table 3: Performance comparison in low-data regimes. The learning difficulty increases progressively from $\mathcal{D}_1$ to $\mathcal{D}_4$, with $\mathcal{D}_1/\mathcal{D}_2$ containing 280 trajectories each and $\mathcal{D}_3/\mathcal{D}_4$ containing 140 each.

| Task (sparse R) | $\mathcal{D}_1$ | | $\mathcal{D}_2$ | | $\mathcal{D}_3$ | | $\mathcal{D}_4$ | |
|------|------|------|------|------|------|------|------|------|
| | QT | RVDT | QT | RVDT | QT | RVDT | QT | RVDT |
| maze2d-umaze | 100.3 | **171.7** | 81.8 | **101.8** | 73.1 | **76.9** | 61.4 | **100.5** |
| maze2d-medium | 137.1 | **187.4** | 175.2 | **190.0** | 163.2 | **182.0** | 98.3 | **175.3** |
| maze2d-large | 109.5 | **140.4** | 81.5 | **90.1** | 104.4 | **131.3** | 100.2 | 95.1 |
| **Average** | 115.6 | **166.5** | 112.8 | **127.3** | 113.6 | **130.1** | 86.6 | **123.6** |

enhances its ability to assemble optimal paths from fragmented sub-trajectories, thereby significantly outperforming all baselines. The AntMaze tasks, characterized by abundant suboptimal and unsuccessful trajectories, pose even greater challenges due to multi-action per state scenarios. While RVDT maintains competitive performance on smaller mazes (`antmaze-u`), virtually all CSM and value-regularized SL methods fail in larger, more complex mazes (`antmaze-m`, `antmaze-l`). In contrast, D-QL, leveraging expressive diffusion policies, achieves the best performance on these challenging tasks. This suggests that highly expressive policies are particularly effective in multi-action per state environments, highlighting the limitations in expressiveness of current CSM methods.

## 5.2 Performance under Imbalanced Trajectory Distributions

**Effectiveness of Expert Data Utilization.** We evaluate whether RVDT, integrating both explicit and implicit return rebalancing mechanisms, can more effectively leverage expert-level data from suboptimal datasets compared to QT and BC. We construct a base dataset by merging the `-m` and `-m-r` datasets, incrementally augmenting it with top-performing trajectories from the `-m-e` dataset. Table 2 shows the performance of BC, QT, and RVDT at varying expert ratios. The results indicate: (i) RVDT consistently outperforms QT and BC at all expert ratios, highlighting the advantage of combining explicit and implicit rebalancing mechanisms compared to using only one (QT) or none (BC) of them; (ii) as the proportion of expert trajectories increases, RVDT and QT show steady improvements, while BC plateaus beyond approximately 30% expert data. This underscores the effectiveness of rebalancing mechanisms in enabling CSM policies to effectively utilize expert data.

**Performance in Low-Data Regimes.** To assess RVDT's performance in low-data scenarios [46], we construct four datasets ($\mathcal{D}_1$-$\mathcal{D}_4$) by subsampling the original `maze2d` datasets. The learning difficulty increases progressively from $\mathcal{D}_1$ to $\mathcal{D}_4$, with $\mathcal{D}_1$ and $\mathcal{D}_2$ containing 280 trajectories each, and $\mathcal{D}_3$ and $\mathcal{D}_4$ containing 140 trajectories. Within each pair, $\mathcal{D}_2$ includes fewer expert-level trajectories than $\mathcal{D}_1$, as does $\mathcal{D}_4$ compared to $\mathcal{D}_3$. Compared to the original 2000-trajectory `maze2d` dataset, these subsets better reflect algorithmic performance in small-sample conditions (details for $\mathcal{D}_1$-$\mathcal{D}_4$ are provided in Appendix F.1). Table 3 shows RVDT consistently outperforming QT, often by a large margin, with average improvements of 45.4%, 14.5%, 14.2%, and 45.6% across $\mathcal{D}_1$-$\mathcal{D}_4$. This substantial performance gain highlights RVDT's superior ability to utilize useful information from limited and low-quality datasets (detailed comparisons in Appendix F.1).

**Stitching Ability.** Maze2D is commonly used to evaluate the trajectory stitching capability of offline RL algorithms [25], as agents are required to compose successful trajectories by integrating fragmented experience. We test RVDT and several baselines across four Maze2D environments

Table 4: Performance comparison on Maze2D environments with sparse and dense rewards.

| | Dataset | CQL | DT | QDT | QT | RVDT |
|---|---|---|---|---|---|---|
| **Sparse R** | maze2d-open-v0 | $216.7 \pm 80.7$ | $196.4 \pm 39.6$ | $190.1 \pm 37.8$ | $497.9 \pm 12.3$ | $\mathbf{634.6} \pm 12.3$ |
| | maze2d-umaze-v1 | $94.7 \pm 23.1$ | $31.0 \pm 21.3$ | $57.3 \pm 8.2$ | $105.4 \pm 4.8$ | $\mathbf{145.1} \pm 3.8$ |
| | maze2d-medium-v1 | $41.8 \pm 13.6$ | $8.2 \pm 4.4$ | $13.3 \pm 5.6$ | $172.0 \pm 6.2$ | $\mathbf{183.5} \pm 4.5$ |
| | maze2d-large-v1 | $49.6 \pm 8.4$ | $2.3 \pm 0.9$ | $31.0 \pm 19.8$ | $240.1 \pm 2.5$ | $\mathbf{254.3} \pm 4.6$ |
| **Dense R** | maze2d-open-v0 | $307.6 \pm 43.5$ | $346.2 \pm 14.3$ | $325.7 \pm 61.4$ | $608.4 \pm 1.9$ | $\mathbf{663.9} \pm 15.9$ |
| | maze2d-umaze-v1 | $72.7 \pm 10.1$ | $-6.8 \pm 10.9$ | $58.6 \pm 3.3$ | $\mathbf{103.1} \pm 7.8$ | $99.5 \pm 4.3$ |
| | maze2d-medium-v1 | $70.9 \pm 9.2$ | $31.5 \pm 3.7$ | $42.3 \pm 7.1$ | $111.9 \pm 1.9$ | $\mathbf{126.9} \pm 8.7$ |
| | maze2d-large-v1 | $90.9 \pm 19.4$ | $45.3 \pm 11.2$ | $62.2 \pm 9.9$ | $177.2 \pm 7.8$ | $\mathbf{197.9} \pm 2.0$ |

Table 5: Component-level breakdown across ablation variants.

| Component | DT | DT-Dup | RDT | QT | VDT | RVDT-Dup | RVDT-Determ | **RVDT** |
|---|---|---|---|---|---|---|---|---|
| Explicit Rebal. | None | Dup. | KL | None | None | Dup. | KL | KL |
| Implicit Rebal. | None | None | None | Q-value | Q-value | Q-value | Q-value | Q-value |
| Policy Type | Determ. | Stoch. | Stoch. | Determ. | Stoch. | Stoch. | Determ. | Stoch. |

Table 6: Performance comparison of ablation variants across MuJoCo tasks.

| Task | DT | DT-Dup | RDT | QT | VDT | RVDT-Dup | RVDT-Determ | **RVDT** |
|---|---|---|---|---|---|---|---|---|
| halfcheetah | 84.2 | 90.3 | 90.5 | 91.2 | 89.5 | 93.4 | 91.5 | **94.9** |
| hopper | 109.5 | 112.1 | 111.9 | 112.3 | 112.6 | 112.1 | 113.6 | **113.8** |
| walker2d | 108.2 | 108.8 | 109.7 | 113.2 | 110.3 | 110.9 | 113.1 | **118.7** |
| **Average** | 100.6 | 103.7 | 104.0 | 105.6 | 104.1 | 105.5 | 106.1 | **109.1** |

with increasing complexity (`open`, `umaze`, `medium`, `large`) under both sparse and dense reward settings. Results in Table 4 show RVDT significantly outperforming baselines across almost all configurations. Notably, RVDT achieves significant improvements over QT under the sparse reward setting, where effective trajectory stitching is crucial. Results demonstrate that integrating explicit and implicit return rebalancing mechanisms substantially improves CSM-based policies' stitching capabilities, particularly for tasks requiring long-horizon credit assignment ability.

### 5.3 Ablation Studies

To quantify the contribution of each individual component in RVDT, we perform ablation studies on merged datasets (`-m-e`, `-m-r`, and `-m`) for MuJoCo continuous control tasks. These merged datasets alleviate coverage limitations in state, action, and return spaces, providing a clearer evaluation of component impacts without dataset-boundary biases [6, 68]. The ablation variants focus on three key aspects: *explicit rebalancing mechanisms*, *implicit rebalancing mechanisms*, and *policy types*. Explicit rebalancing mechanisms include expert-policy KL regularization and expert-level data duplication, while the implicit rebalancing mechanism corresponds to Q-value guidance. For policy types, we consider both Gaussian stochastic policies and deterministic policies. The ablation variants are summarized in Table 5. Specifically, the two major regularizers introduced in our framework, the policy regularization term and the value-based guidance term, correspond to the "Explicit Rebal." (Explicit Rebalancing Mechanism) and "Implicit Rebal." (Implicit Rebalancing Mechanism) abbreviated in Table 6, respectively. The variant RDT denotes the ablated version of RVDT without the value-based guidance component, while VDT represents the version without the policy regularization term. For the ablated variant employing deterministic policies (RVDT-Determ), the KL regularization term is replaced with an MSE objective. In variants where data duplication is applied, the top 50% of trajectories are directly duplicated and appended to the original dataset. Details are provided in Appendix C.

Table 6 summarizes the ablation results and reveals several key insights: (i) RVDT consistently achieves top performance across all tasks, demonstrating the synergistic effectiveness of integrating all proposed components. (ii) Both RDT (explicit rebalancing) and QT/VDT (implicit rebalancing) individually outperform DT, confirming the effectiveness of applying return-coverage rebalanc-

ing mechanisms on DT, and indicating that the combination of rebalancing mechanisms in RVDT provides additional performance gains. (iii) RVDT-Dup, which employs direct expert-level data duplication, achieves comparable or superior performance relative to VDT, supporting our theoretical insights regarding the benefits of explicit rebalancing mechanisms. (iv) A comparison between RVDT-Determ (deterministic policy) and RVDT (stochastic policy) shows that RVDT with stochastic policy outperforms its deterministic counterpart, highlighting the importance of policy expressiveness, particularly in multi-action per state scenarios.

# 6 Conclusion

This work establishes that both expert-level and full-spectrum return-coverage are critical determinants of the performance and sample efficiency of CSM policies in offline RL. Theoretical bounds that quantify the impact of these two aspects of return-coverage on policy optimality and sample complexity are derived, offering a principled understanding of the underlying mechanisms that drive the effectiveness of return rebalancing strategies in CSM-based methods. Moreover, we propose RVDT, the first CSM-based approach that elegantly integrates explicit and implicit return rebalancing mechanisms to effectively alleviate challenges posed by suboptimal datasets. Empirical evaluations on D4RL benchmarks demonstrate that RVDT enhances state-of-the-art performance in offline RL, validating the benefits and efficacy of return-coverage rebalancing.

## Acknowledgements

This work was supported in part by the National Natural Science Foundation of China: 62206248.

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

# A Proofs

## A.1 Proof of Theorem 1

The original proof is provided in [6]. For completeness, we restate the proof within our context to better align with the assumptions and notations adopted in this paper.

*Proof.* Let $g(s_1, a_{1:H})$ be the value of the return by rolling out the open loop sequence of actions $a_{1:H}$ under the deterministic dynamics induced by $T$ and $r$. Then we can write

$$
\begin{aligned}
\mathbb{E}_{s_1}\left[f(s_1)\right] - J(\pi_f) &= \mathbb{E}_{s_1}\left[\mathbb{E}_{\pi_f|s_1}\left[f(s_1) - g_1\right]\right] \\
&= \mathbb{E}_{s_1}\left[\mathbb{E}_{a_{1:H}\sim\pi_f|s_1}\left[f(s_1) - g(s_1, a_{1:H})\right]\right] \\
&\quad + \mathbb{E}_{s_1}\left[\mathbb{E}_{a_{1:H}\sim\pi_f|s_1}\left[g(s_1, a_{1:H}) - g_1\right]\right] \\
&\leq \mathbb{E}_{s_1}\left[\mathbb{E}_{a_{1:H}\sim\pi_f|s_1}\left[f(s_1) - g(s_1, a_{1:H})\right]\right] + \epsilon H^2 \qquad (8)
\end{aligned}
$$

where the last step follows by bounding the magnitude of the difference between $g_1$ and $g(s_1, a_{1:H})$ by $H$ and applying a union bound over the $H$ steps in the trajectory (using the near determinism assumption), namely:

$$
H \cdot \sup_{s_1} \bigcup_t P_{a_t \sim \pi_f|s_1}\left(r_t \neq r(s_t, a_t) \text{ or } s_{t+1} \neq T(s_t, a_t)\right) \leq \epsilon H^2
$$

Now we consider the first term from (8). Again bounding the magnitude of the difference by $H$ we get that

$$
\mathbb{E}_{s_1}\left[\mathbb{E}_{a_{1:H}\sim\pi_f|s_1}\left[f(s_1) - g(s_1, a_{1:H})\right]\right] \leq \mathbb{E}_{s_1}\int_{a_{1:H}} P_{\pi_f}(a_{1:H}\mid s_1)\,\mathbb{I}\left[g(s_1, a_{1:H}) \neq f(s_1)\right]H
$$
(9)

To simplify notation, let $\bar{s}_t = T(s_1, a_{1:t-1})$ be the result of following the deterministic dynamics defined by $T$ up until step $t$. Expanding it out, applying the near determinism, the consistency of $f$, the coverage assumption, canceling some terms, and then inducting we see that:

$$
\begin{aligned}
P_{\pi_f}(a_{1:H}|s_1) &= \pi_f(a_1|s_1)\int_{s_2} P(s_2|s_1, a_1)\, P_{\pi_f}(a_{2:H}|s_1, s_2) \\
&\leq \pi_f(a_1|s_1)\, P_{\pi_f}(a_{2:H}|s_1, \bar{s}_2) + \epsilon \\
&= \pi_\beta(a_1|s_1)\frac{P_{\pi_\beta}(g_1 = f(s_1)|s_1, a_1)}{P_{\pi_\beta}(g_1 = f(s_1)|s_1)}P_{\pi_f}(a_{2:H}|s_1, \bar{s}_2) + \epsilon \\
&\leq \pi_\beta(a_1|s_1)\frac{\epsilon + P_{\pi_\beta}(g_1 - r(s_1, a_1) = f(s_1) - r(s_1, a_1)|s_1, a_1, \bar{s}_2)}{P_{\pi_\beta}(g_1 = f(s_1)|s_1)}P_{\pi_f}(a_{2:H}|s_1, \bar{s}_2) + \epsilon \\
&= \pi_\beta(a_1|s_1)\frac{\epsilon + P_{\pi_\beta}(g_2 = f(\bar{s}_2)|\bar{s}_2)}{P_{\pi_\beta}(g_1 = f(s_1)|s_1)}P_{\pi_f}(a_{2:H}|s_1, \bar{s}_2) + \epsilon \\
&\leq \pi_\beta(a_1|s_1)\frac{P_{\pi_\beta}(g_2 = f(\bar{s}_2)|\bar{s}_2)}{P_{\pi_\beta}(g_1 = f(s_1)|s_1)}P_{\pi_f}(a_{2:H}|s_1, \bar{s}_2) + \epsilon\left(\frac{1}{\alpha_f} + 1\right) \\
&\leq \pi_\beta(a_1|s_1)\pi_\beta(a_2|\bar{s}_2)\frac{P_{\pi_\beta}(g_2 = f(\bar{s}_2)|\bar{s}_2)}{P_{\pi_\beta}(g_1 = f(s_1)|s_1)}\cdot\frac{P_{\pi_\beta}(g_3 = f(\bar{s}_3)|\bar{s}_3, a_3)}{P_{\pi_\beta}(g_2 = f(\bar{s}_2)|\bar{s}_2)}P_{\pi_f}(a_{3:H}|s_1, \bar{s}_3)\Bigg) \\
&\quad + 2\epsilon\left(\frac{1}{\alpha_f} + 1\right) \\
&\leq \prod_{t=1}^{H}\pi_\beta(a_t|\bar{s}_t)\frac{P_{\pi_\beta}(g_H = f(\bar{s}_H)|\bar{s}_H, a_h)}{P_{\pi_\beta}(g_1 = f(s_1)|s_1)} + H\epsilon\left(\frac{1}{\alpha_f} + 1\right) \\
&= \prod_{t=1}^{H}\pi_\beta(a_t|\bar{s}_t)\frac{\mathbb{I}\left[g(s_1, a_{1:H}) = f(s_1)\right]}{P_{\pi_\beta}(g_1 = f(s_1)|s_1)} + H\epsilon\left(\frac{1}{\alpha_f} + 1\right)
\end{aligned}
$$

where the last step follows from the determinism of the trajectory that determines $\bar{s}_H$ and the consistency of $f$. Plugging this back into (9) and noticing that the two indicator functions can never both be 1, we get that:

$$\mathbb{E}_{s_1}\left[\mathbb{E}_{a_{1:H}\sim\pi_f|s_1}\left[f\left(s_1\right)-g\left(s_1,a_{1:H}\right)\right]\right] \leq H^2\epsilon\left(\frac{1}{\alpha_f}+1\right)$$

Plugging this back into (8) yields the result. □

### A.2 Proof of Theorem 2 (performance gap)

We first give the following Lemmas to help the proof of Theorem 2.

**Lemma 1** (Brandfonbrener et al. [6]). *Consider the assumptions of Theorem 1. There exists an optimal conditioning function $f^*$, defined such that $f^*(s_1)$ corresponds to the return of the deterministic optimal policy $\pi^*$ for the underlying MDP. Assume optimal return-coverage $P_{\pi_\beta}\left(g=f^*(s_1)|s_1\right)\geq \alpha_f^*$ holds for all initial states $s_1$. Then we have that*

$$J(\pi^*) - J(\pi_{f^*}^{CSM}) \leq \left(\frac{1}{\alpha_{f^*}}+3\right)H^2\epsilon.$$

*Proof.* Since $f^*(s_1)$ corresponds to the return of the deterministic optimal policy $\pi^*$ for the underlying MDP, so that $\mathbb{E}\left[f^*\left(s_1\right)\right]$ is approximately $J\left(\pi^*\right)$.

Let $T^{\pi^*}\left(s_1,t\right)$ represent the state reached by running $\pi^*$ from $s_1$ for $t$ steps under the deterministic dynamics defined by $T$. Then:

$$f^*\left(s_1\right) = \sum_{t=1}^{H} r\left(T^{\pi^*}\left(s_1,t\right),\pi^*\left(T^{\pi^*}\left(s_1,t\right)\right)\right)$$

Now we have as in the proof of Theorem 1 that the probability that $g\neq f^*(s)$ is bounded by $\epsilon H$, so that

$$\mathbb{E}_{s_1}\left[f^*\left(s_1\right)\right] - J\left(\pi^*\right) = \mathbb{E}_{s_1}\left[\mathbb{E}_{g\sim\pi^*|s_1}\left[f^*\left(s_1\right)-g\right]\right] \leq \mathbb{E}_{s_1}\left[P_{\pi^*}\left(g\neq f^*\left(s_1\right)\mid s_1\right)\cdot H\right] \leq \epsilon H^2$$

Combining this with Theorem 1 yields the result.

□

**Lemma 2** (Achiam et al. [1]). *Let $d_\pi$ refer to the marginal distribution of $P_\pi$ over states only. For any two policies $\pi,\pi'$ we have:*

$$\|d_\pi - d_{\pi'}\|_1 \leq 2H\cdot\mathbb{E}_{s\sim d_\pi}\left[TV\left(\pi(\cdot\mid s)\|\hat{\pi}'(\cdot\mid s)\right)\right]$$

*Proof.* First we will define a few useful objects. Let $d_\pi^h(s) = P_\pi\left(s_h=s\right)$. Let $\Delta_h = \|d_\pi^h(s)-d_{\pi'}^h(s)\|_1$. Let $\delta_h = 2\mathbb{E}_{s\sim d_\pi^h}\left[TV\left(\pi(\cdot\mid s)\|\hat{\pi}'(\cdot\mid s)\right)\right]$.

Now we claim that $\Delta_h \leq \delta_{h-1}+\Delta_{h-1}$ for $h>1$ and $\Delta_1=0$.

To see this, consider some fixed $h$. Note that $d_\pi^h(s)=\int_{s'}d_\pi^{h-1}(s')\int_{a'}\pi\left(a'\mid s'\right)P\left(s\mid s',a'\right)$. Then expanding the definitions and adding and subtracting we see that

$$\begin{aligned}
\Delta_h &= \int_s\left|d_\pi^h(s)-d_{\pi'}^h(s)\right| \\
&\leq \int_s\left|\int_{s'}d_\pi^{h-1}\left(s'\right)\int_{a'}\left(\pi\left(a'\mid s'\right)-\pi'\left(a'\mid s'\right)\right)P\left(s\mid s',a'\right)\right| \\
&\quad + \int_s\left|\int_{s'}\left(d_\pi^{h-1}\left(s'\right)-d_{\pi'}^{h-1}\left(s'\right)\right)\int_{a'}\pi'\left(a'\mid s'\right)P\left(s\mid s',a'\right)\right| \\
&\leq 2\mathbb{E}_{s\sim d_\pi^{h-1}}\left[TV\left(\pi(\cdot\mid s)\|\hat{\pi}'(\cdot\mid s)\right)\right]+\left\|d_\pi^{h-1}-d_{\pi'}^{h-1}\right\|_1 = \delta_{h-1}+\Delta_{h-1}
\end{aligned}$$

xxiii

Now applying the claim and the definition of $d_\pi$ we get that

$$\|d_\pi - d_{\pi'}\|_1 \le \frac{1}{H} \sum_{h=1}^{H} \Delta_h \le \frac{1}{H} \sum_{h=1}^{H} \sum_{j=1}^{h-1} \delta_j \le H \frac{1}{H} \sum_{h=1}^{H} \delta_h = 2H \cdot \mathbb{E}_{s \sim d_\pi} \left[ TV \left( \pi(\cdot \mid s) \| \hat{\pi}'(\cdot \mid s) \right) \right]$$

$\square$

Now we are ready to **prove Theorem 2**.

*Proof.* Let $f^*(s_1)$ denote the return of the optimal deterministic policy $\pi^*$ under deterministic dynamics. According to Theorem 1 and Lemma 1, we have:

$$J(\pi^*) - J(\pi_{f^*}^{\text{CSM}}) \le \epsilon \left( \frac{1}{\alpha_{f^*}} + 3 \right) H^2. \tag{10}$$

$$|J(\pi_f^{\text{CSM}}) - J(\pi_{f^*}^{\text{CSM}})| \le |J(\pi_f^{\text{CSM}}) - J(\pi_\beta)| + |J(\pi_{f^*}^{\text{CSM}}) - J(\pi_\beta)|. \tag{11}$$

Denote $\pi_{f'}^{\text{CSM}}$ with an arbitrary conditioning function $f'$ by $\pi_{f'}$. Let $P_{\pi_{f'}}$ and $P_{\pi_\beta}$ denote the stationary distribution over states, actions, and returns induced by policy $\pi_{f'}$ and $\pi_\beta$ respectively.

$$
\begin{aligned}
|J(\pi_{f'}) - J(\pi_\beta)| &= H | \left( \mathbb{E}_{P_{\pi_{f'}}}[r(s,a)] - \mathbb{E}_{P_{\pi_\beta}}[r(s,a)] \right) | \\
&\le H \left\| d_{\pi_{f'}} - d_{\pi_\beta} \right\|_1 \\
&\le 2H^2 \cdot \mathbb{E}_{s \sim d_{\pi_\beta}} \left[ TV \left( \pi_{f'}(\cdot|s) \| \pi_\beta(\cdot|s) \right) \right] \\
&\le H^2 \cdot \mathbb{E}_{s \sim d_{\pi_\beta}} \left[ \int_a |\pi_{f'}(a|s) - \pi_\beta(a|s)| \right] \\
&\le H^2 \mathbb{E}_{s \sim d_{\pi_\beta}} \left[ \int_a \pi_\beta(a|s) \frac{P_{\pi_\beta}(f'(s)|s) - P_{\pi_\beta}(f'(s)|s,a)}{P_{\pi_\beta}(f'(s)|s)} \right] \\
&\le \frac{H^2}{\alpha_{f'}} \mathbb{E}_{s \sim d_{\pi_\beta}} \left[ \int_a \pi_\beta(a|s) \right],
\end{aligned}
\tag{12}
$$

where the third inequality above utilizes the result of Lemma 2.

Denote $\mathbb{E}_{s \sim d_{\pi_\beta}}[\int_a \pi_\beta(a|s)]$ by $\bar{\alpha}_\beta$. By equation (12) and equation (11) we have

$$|J(\pi_f^{\text{CSM}}) - J(\pi_{f^*}^{\text{CSM}})| \le (\frac{1}{\alpha_f} + \frac{1}{\alpha_f^*}) H^2 \bar{\alpha}_\beta \tag{13}$$

Combine (10) and (13) we conclude

$$J(\pi^*) - J(\pi_f^{\text{CSM}}) \le \epsilon(\frac{1}{\alpha_f^*} + 3)H^2 + (\frac{1}{\alpha_f} + \frac{1}{\alpha_f^*})H^2 \bar{\alpha}_\beta \tag{14}$$

Where we have the condition

$$\alpha_f + \alpha_{f^*} \le 1, \alpha_f \in (0,1]; \tag{15}$$
$$\text{if} \quad f \ne f^*, \alpha_f^* \in (0,1]. \tag{16}$$

That is $\beta$ should have both good coverage on sample $f$ and the optimal $f^*$, when expert data increase, $f^*$ will increase but $f$ may decrease.

$$\boxed{\text{Should balance expert data and full-spectrum data!}} \tag{17}$$

$\square$

## A.3 Proof of Theorem 3 (sample complexity)

We first introduce the following useful lemma, whose proof synthesizes the techniques from Achiam et al.[1] and Brandfonbrener et al.[6].

**Lemma 3** (Sample complexity of CSM). *Consider any conditioning function $f : \mathcal{S} \to \mathbb{R}$ such that the following assumptions hold: (i) Bounded occupancy mismatch: $P_{\pi_f^{CSM}}(s) \leq C_f P_{\pi_\beta}(s)$ for all $s$. (ii) Return-coverage: $P_{\pi_\beta}(g = f(s) \mid s) \geq \alpha_f$ for all $s$. (iii) The policy class $\Pi$ is finite. (iv) $|\log \pi(a \mid s, g) - \log \pi(a' \mid s', g')| \leq c$ for any $(a, s, g, a', s', g')$ and all $\pi \in \Pi$. (v) The approximation error of $\Pi$ is bounded by $\epsilon_{approx}$, i.e., $\min_{\pi \in \Pi} L(\pi) \leq \epsilon_{approx.}$.*

*Define the expected loss for a CSM algorithm as*

$$L(\hat{\pi}) = \mathbb{E}_{s \sim P_{\pi_\beta}} \mathbb{E}_{g \sim P_{\pi_\beta}(\cdot | s)} \left[ \text{KL} \left( P_{\pi_\beta}(\cdot \mid s, g) \| \hat{\pi}(\cdot \mid s, g) \right) \right].$$

*Then with probability at least $1 - \delta$,*

$$J\left(\pi_f^{CSM}\right) - J\left(\hat{\pi}_f\right) \leq O\left( \frac{C_f}{\alpha_f} H^2 \left( \sqrt{c} \left( \frac{\log |\Pi|/\delta}{N} \right)^{1/4} + \sqrt{\epsilon_{approx}} \right) \right). \tag{18}$$

*Proof.* Applying the definition of $J$ and Lemma 2, we get

$$J\left(\pi_f^{\text{CSM}}\right) - J\left(\hat{\pi}_f\right) = H\left( \mathbb{E}_{P_{\pi_f^{\text{CSM}}}}[r(s,a)] - \mathbb{E}_{P_{\hat{\pi}_f}}[r(s,a)] \right)$$

$$\leq H \left\| d_{\pi_f^{\text{CSM}}} - d_{\hat{\pi}_f} \right\|_1$$

$$\leq 2 \cdot \mathbb{E}_{s \sim d_{\pi_f^{\text{CSM}}}} \left[ TV\left( \pi_f^{\text{CSM}}(\cdot \mid s) \| \hat{\pi}_f(\cdot \mid s) \right) \right] H^2$$

Expanding definitions, using the multiply and divide trick, and applying the assumptions:

$$2 \cdot \mathbb{E}_{s \sim d_{\pi_f^{\text{CSM}}}} \left[ TV\left( \pi_f^{\text{CSM}}(\cdot \mid s) \| \hat{\pi}_f(\cdot \mid s) \right) \right]$$

$$= \mathbb{E}_{s \sim d_{\pi_f^{\text{CSM}}}} \left[ \int_a \left| P_{\pi_\beta}(a \mid s, f(s)) - \hat{\pi}(a \mid s, f(s)) \right| \right]$$

$$= \mathbb{E}_{s \sim d_{\pi_f^{\text{CSM}}}} \left[ \frac{P_{\pi_\beta}(f(s) \mid s)}{P_{\pi_\beta}(f(s) \mid s)} \int_a \left| P_{\pi_\beta}(a \mid s, f(s)) - \hat{\pi}(a \mid s, f(s)) \right| \right]$$

$$\leq \frac{C_f}{\alpha_f} \mathbb{E}_{s \sim d_{\pi_\beta}} \left[ P_{\pi_\beta}(f(s) \mid s) \int_a \left| P_{\pi_\beta}(a \mid s, f(s)) - \hat{\pi}(a \mid s, f(s)) \right| \right]$$

$$\leq \frac{C_f}{\alpha_f} \mathbb{E}_{s \sim d_{\pi_\beta}} \left[ \int_g P_{\pi_\beta}(g \mid s) \int_a \left| P_{\pi_\beta}(a \mid s, g) - \hat{\pi}(a \mid s, g) \right| \right]$$

$$= 2 \frac{C_f}{\alpha_f} \mathbb{E}_{s \sim d_{\pi_f^{\text{CSM}}}, g \sim P_{\pi_\beta}|s} \left[ TV\left( P_{\pi_\beta}(\cdot \mid s, g) \| \hat{\pi}(\cdot \mid s, g) \right) \right]$$

$$\leq \frac{C_f}{\alpha_f} \sqrt{2L(\hat{\pi})},$$

where the last step comes from Pinsker's inequality.

Combining with the above bound on the difference in expected values, we can get:

$$J\left(\pi_f^{\text{CSM}}\right) - J\left(\hat{\pi}_f\right) \leq \frac{C_f}{\alpha_f} H^2 \sqrt{2L(\hat{\pi})}. \tag{19}$$

We now write $L(\pi) = \bar{L}(\pi) - H_{\pi_\beta}$, where $H_{\pi_\beta} = -\mathbb{E}_{(s,a,g) \sim P_{\pi_\beta}} \left[ \log P_{\pi_\beta}(a \mid s, g) \right]$ and $\bar{L}(\pi) := -\mathbb{E}_{(s,a,g) \sim P_{\pi_\beta}} [\log \pi(a \mid s, g)]$ is the cross-entropy loss.

Denoting $\pi^\dagger \in \arg \min_{\pi \in \Pi} L(\pi)$, we have

$$L(\hat{\pi}) = L(\hat{\pi}) - L\left(\pi^\dagger\right) + L\left(\pi^\dagger\right) \leq \bar{L}(\hat{\pi}) - \bar{L}\left(\pi^\dagger\right) + \epsilon_{\text{approx}}.$$

Denoting $\hat{L}$ the empirical cross-entropy loss that is minimized by $\hat{\pi}$, we may further decompose

$$\bar{L}(\hat{\pi}) - \bar{L}\left(\pi^{\dagger}\right) = \bar{L}(\hat{\pi}) - \hat{L}(\hat{\pi}) + \hat{L}(\hat{\pi}) - \hat{L}\left(\pi^{\dagger}\right) + \hat{L}\left(\pi^{\dagger}\right) - \bar{L}\left(\pi^{\dagger}\right)$$
$$\leq 2 \sup_{\pi \in \Pi} |\bar{L}(\pi) - \hat{L}(\pi)|$$

Under the assumptions on bounded loss differences, we may bound this, e.g., using McDiarmid's inequality and a union bound on $\Pi$ to obtain the final result in (18) [50, 6]. $\square$

Now we are ready to prove the theorem 3

*Proof.* By Lemma 3 we have the following result that

$$J\left(\pi_f^{CSM}\right) - J\left(\hat{\pi}_f\right) \leq O\left(\frac{C_f}{\alpha_f} H^2 \left(\sqrt{c}\left(\frac{\log |\Pi|/\delta}{N}\right)^{1/4} + \sqrt{\epsilon_{\text{approx}}}\right)\right). \tag{20}$$

By Theorem 2 we have the following result that

$$J(\pi^*) - J(\pi_f^{\text{CSM}}) \leq (\frac{1}{\alpha_f^*} + 3)H^2\epsilon + (\frac{1}{\alpha_f} + \frac{1}{\alpha_f^*})H^2C. \tag{21}$$

Thus, by combining the results from (20) and (21), we conclude that

$$J(\pi^*) - J\left(\hat{\pi}_f\right) \leq O\left(\frac{C_f}{\alpha_f} H^2 \left(\sqrt{c}\left(\frac{\log |\Pi|/\delta}{N}\right)^{1/4} + \sqrt{\epsilon_{\text{approx}}}\right) + \left(\frac{\epsilon + C}{\alpha_f^*} + \frac{C}{\alpha_f}\right) H^2\right).$$

$\square$

## A.4 Proof of Proposition 1

*Proof.* To prove Proposition 1, we show that the KL regularization term in (4) is proportional to the negative log-likelihood (NLL) loss, under the assumption that the policy $\pi_\theta$ is a factorized Gaussian distribution with a fixed isotropic covariance. The equivalence then implies that KL regularization explicitly reweights the NLL loss, thereby implementing an explicit resampling mechanism.

Recall the KL divergence between two multivariate Gaussians $\mathcal{N}(\mu_1, \Sigma_1)$ and $\mathcal{N}(\mu_2, \Sigma_2)$ is given by:

$$D_{\text{KL}}\left(\mathcal{N}(\mu_1, \Sigma_1) \,\|\, \mathcal{N}(\mu_2, \Sigma_2)\right) = \frac{1}{2}\left[\text{tr}(\Sigma_2^{-1}\Sigma_1) + (\mu_2 - \mu_1)^{\top}\Sigma_2^{-1}(\mu_2 - \mu_1) - d + \log\frac{|\Sigma_2|}{|\Sigma_1|}\right], \tag{22}$$

where $d$ is the dimensionality of the action space.

In our setting, the policy is parameterized as $\pi_\theta(a|s) = \mathcal{N}(\mu_\theta(s), \xi I)$ and the target is a Dirac approximation centered at the action $a$, i.e., $\mathcal{N}(a, \xi I)$. Substituting into (22), we obtain:

$$D_{\text{KL}}\left(\mathcal{N}(\mu_\theta(s), \xi I) \,\|\, \mathcal{N}(a, \xi I)\right) = \frac{1}{2}\left[\text{tr}(I) + (\mu_\theta(s) - a)^{\top}(\xi I)^{-1}(\mu_\theta(s) - a) - d + \log 1\right],$$

where we use the fact that $\Sigma_1 = \Sigma_2 = \xi I$. Simplifying yields:

$$D_{\text{KL}}\left(\mathcal{N}(\mu_\theta(s), \xi I) \,\|\, \mathcal{N}(a, \xi I)\right) = \frac{1}{2\xi}\|\mu_\theta(s) - a\|^2.$$

Taking expectation over $(s, a) \sim \mathcal{D}$ gives:

$$\mathbb{E}_{(s,a)\sim\mathcal{D}}\left[D_{\text{KL}}\left(\mathcal{N}(\mu_\theta(s), \xi I) \,\|\, \mathcal{N}(a, \xi I)\right)\right] = \frac{1}{2\xi}\,\mathbb{E}_{(s,a)\sim\mathcal{D}}\left[\|\mu_\theta(s) - a\|^2\right],$$

which shows that the KL divergence is proportional to the mean squared error (MSE) loss between $\mu_\theta(s)$ and the ground-truth action $a$.

The log-likelihood of the Gaussian policy $\pi_\theta(a|s) = \mathcal{N}(a; \mu_\theta(s), \xi I)$ is:

$$\log \pi_\theta(a|s) = -\frac{d}{2} \log(2\pi\xi) - \frac{1}{2\xi} \|a - \mu_\theta(s)\|^2.$$

Taking the negative log-likelihood gives:

$$-\log \pi_\theta(a|s) = \frac{1}{2\xi} \|\mu_\theta(s) - a\|^2 + \frac{d}{2} \log(2\pi\xi).$$

The constant term $\frac{d}{2} \log(2\pi\xi)$ is independent of the policy parameters $\theta$ and does not affect optimization. Therefore, minimizing the KL divergence is equivalent (up to scaling and additive constants) to minimizing the NLL loss:

$$\mathbb{E}_{(s,a)\sim\mathcal{D}} \left[ D_{\mathrm{KL}} \left( \mathcal{N}(\mu_\theta(s), \xi I) \,\|\, \mathcal{N}(a, \xi I) \right) \right] = \mathbb{E}_{(s,a)\sim\mathcal{D}} \left[ -\log \pi_\theta(a|s) \right] + \text{constant}. \tag{23}$$

Recall that the RDT objective in (4) augments the standard NLL loss with a KL regularization term over expert trajectories:

$$\mathcal{L}_{\mathrm{RDT}}(\theta) = \mathbb{E}_{\tau\sim\mathcal{D}} \sum_{i=1}^{H} \left[ -\log \pi_\theta(a_i|s_i, g(\tau_i), \bar{\tau}_{t-1}^K) \right] + \alpha \, \mathbb{E}_{\tau\sim\mathcal{D}_e} \sum_{i=1}^{H} \left[ D_{\mathrm{KL}} \left( \pi_\theta(\cdot|s_i, g(\tau_i), \bar{\tau}_{t-1}^K) \,\|\, \pi_e(\cdot|s_i) \right) \right].$$

If $\pi_e$ is a delta function centered at $a_i$ (i.e., using the empirical action from the expert trajectory), and both policies are Gaussians with identical covariance $\xi I$, the KL term simplifies to an MSE loss, and hence to an NLL term by (23). Thus, the second expectation term becomes equivalent to an additional NLL loss over $\mathcal{D}_e$ with weighting factor $\alpha$.

This yields the final objective:

$$\mathcal{L}_{\mathrm{RDT}}(\theta) = \mathbb{E}_{\tau\sim\mathcal{D}} \left[ (1 + \alpha \cdot \mathbb{I}[\tau \in \mathcal{D}_e]) \cdot \sum_{i=1}^{H} -\log \pi_\theta(a_i|s_i, g(\tau_i), \bar{\tau}_{t-1}^K) \right] + \text{constant}.$$

Since the constant term does not affect the optimization process, the proof is thus completed. $\quad\square$

## A.5 Formal Statement and Proof of Proposition 2

*Proof.* To formally state Proposition 2, we first define the loss functions of the respective algorithms.

We begin by explicitly stating our key assumptions:

- Infinite dataset size, i.e., $|D| \to \infty$
- Both $\pi$ and $\pi_\beta$ are gaussian policy

Under these conditions, it has been shown that minimizing the negative log-likelihood is equivalent to minimizing the KL divergence between $\pi$ and $\pi_\beta$:

$$\begin{aligned} \mathrm{NLL} &= E_{(s,a)\sim D}[-\log \pi(a|s)] \\ &= E_{s\sim d_{\pi_\beta}(\cdot)}[D_{KL}(\pi(\cdot|s)\|\pi_\beta(\cdot|s))] \end{aligned} \tag{24}$$

As shown in Proposition 1, KL regularization corresponds to a reweighting of the data distribution. We define the reweighted dataset $\tilde{D}$ and the corresponding mixture policy $\pi_{\tilde\beta}$ as:

$$\begin{aligned} \tilde{D} &= \frac{1}{1+\alpha} D + \frac{\alpha}{1+\alpha} D_e \\ \pi_{\tilde\beta} &= \frac{1}{1+\alpha} \pi_\beta + \frac{\alpha}{1+\alpha} \pi_e \end{aligned} \tag{25}$$

With these definitions, we rewrite the learning objectives for DT, QT, RDT and RVDT:

$$\pi_{\mathrm{dt}}^* := \arg\max_\pi L_{\mathrm{DT}}(\pi) = \arg\max_\pi \mathbb{E}_{s\sim d_{\pi_\beta}(\cdot)}[-D_{KL}(\pi(\cdot|s)\|\pi_\beta(\cdot|s))] = \pi_\beta \tag{26}$$

$$\pi_{\text{qt}}^* := \arg\max_\pi L_{\text{QT}}(\pi) = \arg\max_\pi \mathbb{E}_{s\sim d_{\pi_\beta}(\cdot)}[-D_{KL}(\pi(\cdot|s)\|\pi_\beta(\cdot|s)) + \eta\mathbb{E}_{a\sim\pi(\cdot|s)}[Q_\pi(s,a)]] \tag{27}$$

$$\pi_\omega^* := \arg\max_\pi L_{\text{RDT}}(\pi) = \arg\max_\pi \mathbb{E}_{s\sim d_{\pi_\beta}(\cdot)}[-D_{KL}(\pi(\cdot|s)\|\pi_{\tilde\beta}(\cdot|s))] = \pi_{\tilde\beta} \tag{28}$$

$$\pi_e^* := \arg\max_\pi L_{\text{RVDT}}(\pi) = \arg\max_\pi \mathbb{E}_{s\sim d_{\pi_\beta}(\cdot)}[-D_{KL}(\pi(\cdot|s)\|\pi_{\tilde\beta}(\cdot|s)) + \eta\mathbb{E}_{a\sim\pi(\cdot|s)}[Q_\pi(s,a)]] \tag{29}$$

Based on these optimization objectives, we can formally restate Proposition 2.

**Proposition 3** (Formal statement of Proposition 2). *Let $\pi_e^*$, $\pi_w^*$, $\pi_{qt}^*$, and $\pi_{dt}^*$ denote the optimal policies learned from the objectives of RVDT (29), RDT (28), QT (27) and DT (26) respectively. Given the behavior policy $\pi_\beta$, there exists an expert-level subset $\mathcal{D}_e \subseteq \mathcal{D}$ and an appropriate coefficient $\alpha \geq 0$ such that the following inequalities hold for all states $s \sim d_{\pi_\beta}$:*

$$\mathbb{E}_s V^{\pi_e^*}(s) \geq \mathbb{E}_s V^{\pi_w^*}(s) \geq \mathbb{E}_s V^{\pi_{dt}^*}(s), \quad and$$
$$\mathbb{E}_s V^{\pi_e^*}(s) \geq \mathbb{E}_s V^{\pi_{qt}^*}(s) \geq \mathbb{E}_s V^{\pi_{dt}^*}(s). \tag{30}$$

Since $\pi_{\text{dt}}^* = \pi_\beta$, we aim to prove the following policy improvement results:

$$\mathbb{E}_{s\sim d_{\pi_\beta}(\cdot)} V^{\pi_\beta}(s) \leq \mathbb{E}_{s\sim d_{\pi_\beta}(\cdot)} V^{\pi_\omega^*}(s) \tag{31}$$

$$\mathbb{E}_{s\sim d_{\pi_\beta}(\cdot)} V^{\pi_{\text{qt}}^*}(s) \leq \mathbb{E}_{s\sim d_{\pi_\beta}(\cdot)} V^{\pi_e^*}(s) \tag{32}$$

$$\mathbb{E}_{s\sim d_{\pi_\beta}(\cdot)} V^{\pi_\beta}(s) \leq \mathbb{E}_{s\sim d_{\pi_\beta}(\cdot)} V^{\pi_{\text{qt}}}(s) \tag{33}$$

$$\mathbb{E}_{s\sim d_{\pi_\beta}(\cdot)} V^{\pi_\omega^*}(s) \leq \mathbb{E}_{s\sim d_{\pi_\beta}(\cdot)} V^{\pi_e^*}(s) \tag{34}$$

Inequalities (31) and (32) follow directly since (1) and (6) are special cases of (4) and (37), respectively, when $\alpha = 0$.

We now prove (33).

$$\begin{aligned}
&\eta\mathbb{E}_{s\sim d_{\pi_\beta}(\cdot)}[V^{\pi_{\text{qt}}^*}(s)] \\
&= \eta\mathbb{E}_{s\sim d_{\pi_\beta}(\cdot),a\sim\pi_{\text{qt}}^*(\cdot|s)}[Q^{\pi_{\text{qt}}^*}(s,a)] \\
&\geq \eta\mathbb{E}_{s\sim d_{\pi_\beta}(\cdot),a\sim\pi_{\text{qt}}^*(\cdot|s)}[Q^{\pi_{\text{qt}}^*}(s,a)] - \mathbb{E}_{s\sim d_{\pi_\beta}(\cdot)}[D_{KL}(\pi(\cdot|s)\|\pi_\beta(\cdot|s))] \\
&\geq \eta\mathbb{E}_{s\sim d_{\pi_\beta}(\cdot),a\sim\pi_\beta}[Q^{\pi_\beta}(s,a)] - \mathbb{E}_{s\sim d_{\pi_\beta}(\cdot)}[D_{KL}(\pi_\beta(\cdot|s)\|\pi_\beta(\cdot|s))] \\
&= \eta\mathbb{E}_{s\sim d_{\pi_\beta}(\cdot),a\sim\pi_\beta}[Q^{\pi_\beta}(s,a)] = \eta\mathbb{E}_{s\sim d_{\pi_\beta}(\cdot)}[V^{\pi_\beta}(s)]
\end{aligned} \tag{35}$$

where the first inequality follows the non-negativity of KL divergence, the second inequality follows $\pi_{\text{qt}}^*$ is the optimal policy of 27.

Similarly, for (34):

$$\begin{aligned}
&\eta\mathbb{E}_{s\sim d_{\pi_\beta}(\cdot)}[V^{\pi_e^*}(s)] \\
&= \eta\mathbb{E}_{s\sim d_{\pi_\beta}(\cdot),a\sim\pi_e^*(\cdot|s)}[Q^{\pi_e^*}(s,a)] \\
&\geq \eta\mathbb{E}_{s\sim d_{\pi_\beta}(\cdot),a\sim\pi_e^*(\cdot|s)}[Q^{\pi_e^*}(s,a)] - \mathbb{E}_{s\sim d_{\pi_\beta}(\cdot)}[D_{KL}(\pi(\cdot|s)\|\pi_{\tilde\beta}(\cdot|s))] \\
&\geq \eta\mathbb{E}_{s\sim d_{\pi_\beta}(\cdot),a\sim\pi_{\tilde\beta}}[Q^{\pi_{\tilde\beta}}(s,a)] = \eta\mathbb{E}_{s\sim d_{\pi_\beta}(\cdot)}[V^{\pi_{\tilde\beta}}(s)]
\end{aligned} \tag{36}$$

This completes the proof. $\qquad\square$

# B  Training and Inference

We instantiate the previously described formulation using a transformer-based architecture. The RVDT architecture extends the DT framework by explicitly accounting for stochastic policies, predicting the policy's mean and log-variance through two separate fully connected layers. The comprehensive training procedure for RVDT is summarized in Algorithm 1. For notational simplicity, Algorithm 1 represents the policy input using the shorthand $\pi_\theta(\cdot|s_{i-K:i}, a_{i-K:i-1}, g_{i-K:i})$, which explicitly treats the current state–return pair $(s_i, g_i)$ as part of the context window. When $i - K < 0$, the context window is truncated to the available prefix of the trajectory.

In scenarios without supplementary datasets or direct access to expert policies, we rely on cumulative returns to differentiate trajectory quality. Specifically, we select the top $\rho\%$ trajectories ($\rho$ is a hyperparameter), ranked by cumulative return, to form the expert dataset $\mathcal{D}_e$. The expert policy $\pi^e$ is subsequently trained via behavior cloning on $\mathcal{D}_e$. Note that alternative imitation learning approaches such as GAIL [20] or IRL [3] could similarly be utilized to derive $\pi^e$.

RVDT employs a two-step sampling process to ensure uniform sampling of sub-trajectories of length $K$ from the replay buffer $\mathcal{T}_{\text{replay}}$. Initially, an entire trajectory is sampled with probability proportional to its length, and subsequently, a sub-trajectory of length $K$ is uniformly extracted. For environments characterized by non-negative dense rewards, this approach resembles an importance sampling scheme [67].

We leverage offline Dynamic Programming techniques to train the value network $Q_\psi$, optimizing via Temporal Difference (TD) learning [53]. In practical implementation, standard stabilization techniques such as double Q-learning [18] and the use of a target network [43] are adopted. It is noteworthy that in the offline RL setting, conservative Q-learning (CQL) [40] is typically employed to mitigate the out-of-distribution issue by maintaining policy similarity to the behavior policy. However, since the RVDT inherently incorporates behavior cloning, which naturally ensures similarity to the behavior policy, the additional adoption of CQL yields minimal performance gains.

During inference, return-conditioned supervised methods initially require specifying a target return as the first RTG token, subsequently subtracting sampled rewards to update the RTG token iteratively. According to Theorem 1, achieving the target return critically depends on the coverage of the specified return in the dataset. Hence, a practical choice is to utilize the maximum return observed in the training dataset as the initial target. Nonetheless, given RVDT's integration with dynamic programming principles and its strong trajectory stitching capability, the actual optimal target return may surpass the dataset's maximum observed return. Building upon the inference strategy of QT, we randomly sample candidate returns within a positive $10\%$ margin above the maximum return observed in the dataset, thereby obtaining multiple candidate return-to-go tokens $\hat{g}_0^0, \hat{g}_0^1, \ldots, \hat{g}_0^m$. Actions corresponding to each candidate return-to-go token are simultaneously generated. Subsequently, we employ the learned Q-value function to preferentially select actions associated with higher returns. This selection mechanism is formally expressed as: $\hat{a}_t^i = \arg\max_{\hat{a}_t^i} Q_\psi(s_t, \hat{a}_t^i)$, where $\hat{a}_t^i \sim \pi(\cdot|s_t, g_t^i, \bar{\tau}_{t-1}^K)$.

# C  Experimental Details

## C.1  Benchmark Details

Our evaluation is performed across five diverse domains in the D4RL benchmarks [16]: **Gym**, **Adroit**, **Kitchen**, **Maze2D**, and **AntMaze**. These domains are designed to comprehensively evaluate the capability of offline RL algorithms in handling different types of dynamics, reward structures, and state-action distributions.

- **Gym-MuJoCo** locomotion tasks serve as standard benchmarks for continuous control. These tasks, such as HalfCheetah, Hopper, and Walker2d, are characterized by smooth reward functions and dense feedback signals. The datasets include trajectories generated by policies of varying quality: random, medium, expert, and medium-expert, representing different levels of optimality. While expert and medium-expert provide high-quality trajectories, medium, random, and medium-replay contain suboptimal and exploratory behavior, offering diverse state-action coverage for offline learning.

---

**Algorithm 1:** RVDT: Return-rebalanced Value-regularized Decision Transformer

---

**Input:** Sequence length $K$, dataset $\mathcal{D}$, hyperparameters $\alpha, \eta$, expert ratio $\rho\%$
**Initialize:** Policy $\pi_\theta$, critic $Q_\psi$
Construct expert dataset $\mathcal{D}_e$ by selecting top $\rho\%$ trajectories from $\mathcal{D}$;
Train expert policy $\pi^e$ on $\mathcal{D}_e$ via BC;
**for** $t = 1$ **to** $T$ **do**

    Sample a trajectory $\tau \sim \mathcal{D}$ and a length-$K$ subsequence from $\tau$;
    Denote the sampled subsequence as $\{(s_i, a_i, g_i)\}_{i=1}^K$ by re-indexing;
    `// Critic update`
    Update critic $Q_\psi$ using offline TD learning on transitions sampled from $\mathcal{D}$;
    `// Policy update`
    **for** $i = 1$ **to** $K$ **do**

        Sample action $\bar{a}_i \sim \pi_\theta(\cdot|s_{i-K:i}, a_{i-K:i-1}, g_{i-K:i})$;
        Compute $Q_\psi(s_i, \bar{a}_i)$;
        Compute $\text{KL}\big(\pi_\theta(\cdot|s_{i-K:i}, a_{i-K:i-1}, g_{i-K:i}) \,\|\, \pi^e(\cdot|s_i)\big)$;

    Update policy $\pi_\theta$ by minimizing Eq. (7);

**return** $\pi_\theta$;

---

- **Adroit** tasks involve high-dimensional dexterous manipulation using a 24-DoF robotic hand. The datasets are primarily collected from *human demonstrations* and *behavior cloned policies*, resulting in narrow state-action coverage and strong multi-modal behavior. Due to the inherent complexity of the hand's control space, policies must be effectively regularized to avoid out-of-distribution (OOD) actions. The sparse nature of successful demonstrations and the high variance in human trajectories further challenge offline RL algorithms in terms of generalization and stability.

- **Kitchen** environment introduces long-horizon, multi-task manipulation scenarios, where an agent is required to execute a fixed sequence of four subtasks to achieve a goal configuration, such as turning on the microwave, opening the cabinet, or switching on the light. This setting emphasizes the importance of *temporal abstraction* and *multi-task planning*. The datasets are collected from *demonstrator policies* that exhibit varying degrees of task completion, leading to fragmented sub-trajectories that need to be stitched together effectively during offline learning.

- **Maze2D** tasks are specifically designed to evaluate the ability of offline RL algorithms to *stitch together fragmented sub-trajectories* and recover globally optimal paths in a continuous navigation setting. Unlike Gym-MuJoCo, Maze2D environments present *sparse rewards*, requiring the agent to combine distant state transitions to solve navigation tasks effectively. Although Maze2D also provides a dense reward setting, we primarily focus on its sparse reward version throughout this work unless otherwise specified. The agent controls a 2D ball in complex maze-like structures, where the challenge lies in leveraging suboptimal paths to construct optimal global trajectories.

- **AntMaze** represents a more challenging extension of Maze2D by introducing an 8-DoF *Ant* robot instead of a simple 2D ball. The sparse reward structure remains, but the high-dimensional state space and unstable dynamics of the *Ant* make trajectory stitching significantly more difficult. Successful navigation requires effective trajectory stitching and long-horizon credit assignment to traverse disconnected regions of the maze. Datasets are collected using goal-conditioned policies with diverse start-goal configurations, introducing variability in trajectory quality and return distributions.

Dataset descriptions and policy types for representative tasks in the D4RL benchmarks is given in Table 7.

Table 7: Dataset descriptions and policy types for representative tasks in the D4RL benchmarks.

| Task | Policy Type | Description |
|------|-------------|-------------|
| hopper-medium-expert-v2 | Expert + Medium policy | A 50-50 mixture of trajectories from a near-optimal expert policy and a medium-performance policy. |
| hopper-medium-v2 | Medium behavior policy | Collected by a policy trained to partial convergence, representing moderately optimal behavior. |
| hopper-medium-replay-v2 | Medium + Experience replay | A replay buffer from the medium policy, including a wide range of suboptimal and early-stage transitions. |
| pen-human-v1 | Human demonstrations | Trajectories collected via human teleoperation using a virtual reality interface. |
| pen-cloned-v1 | Behavior cloning policy | Trajectories generated by a policy trained via behavioral cloning on human demonstration data. |
| kitchen-complete-v0 | Expert policy | Expert demonstrations accomplishing a fixed sequence of four goal-directed tasks. |
| kitchen-partial-v0 | Expert policy | Expert demonstrations completing a variable subset of kitchen tasks in different orders. |
| maze2d-umaze-v1 | Trajectory tracking (suboptimal) | Random-walk trajectories collected in a small U-shaped maze. |
| maze2d-medium-v1 | Trajectory tracking (suboptimal) | Random-walk trajectories in a medium-sized maze with increased spatial complexity. |
| maze2d-large-v1 | Trajectory tracking (suboptimal) | Random-walk trajectories in the largest maze with long-horizon navigation. |
| antmaze-umaze-v0 | Expert policy | Goal-conditioned expert demonstrations in a small U-shaped maze. |
| antmaze-umaze-diverse-v0 | Expert policy (diverse goals) | Goal-reaching demonstrations in the small maze with diverse start-goal pairs. |
| antmaze-medium-diverse-v0 | Expert policy (diverse goals) | Goal-directed expert trajectories in a medium maze with varying start-goal configurations. |
| antmaze-large-diverse-v0 | Expert policy (sparse + diverse) | Sparse, long-horizon goal-reaching demonstrations in a large maze using an 8-DoF quadruped agent. |

# D  Implementation Details

**The policy network** of RVDT is implemented as a Decision Transformer, built upon the open-source `minGPT` codebase[3] and the official QT implementation[4]. Detailed model parameters are provided in Table 8.

**The expert policy** $\pi^e$ is modeled using a fully connected neural network and trained via behavior cloning. The corresponding model parameters are listed in Table 9.

**The Q-networks** are represented by three-layer MLPs with Mish activations and 256 hidden units for each layer.

All networks are trained using the Adam optimizer [35].

The code of RVDT, including all hyperparameter configurations and experimental setups, is provided in the supplementary material for reproducibility.

The experiments were conducted on a server equipped with two AMD EPYC 7542 32-Core Processors and 8 NVIDIA GeForce RTX 4090 GPUs with 24 GB of memory. For each task, the average training time on a single GPU was approximately 20,000 seconds, with a memory consumption of around 3000MB. These computational settings ensure reproducibility and align with the reported performance metrics.

---

[3] https://github.com/karpathy/minGPT
[4] https://github.com/charleshsc/QT

Table 8: Hyperparameters of RVDT in our experiment.

| Parameter | Value |
|---|---|
| Number of layers | 4 |
| Number of attention heads | 4 |
| Embedding dimension | 256 |
| Nonlinearity function | ReLU |
| Batch size | 256 |
| Context length $K$ | 20 |
| Dropout | 0.1 |
| Learning rate | $3.0\mathrm{e}-4$ |
| Weight decay | $1.0\mathrm{e}-4$ |

Table 9: Hyperparameters of $\pi^e$ in our experiment.

| Parameter | Value |
|---|---|
| Number of layers | 4 |
| Embedding dimension | 256 |
| Nonlinearity function | ReLU |
| Batch size | 256 |
| Dropout | 0.1 |
| Learning rate | $1.0\mathrm{e}-4$ |
| Weight decay | $1.0\mathrm{e}-4$ |

# E  Hyperparameters for RVDT

We recall the loss function for RVDT as follows:

$$
\begin{aligned}
\mathcal{L}_{\mathrm{RVDT}}(\theta) =& \kappa \mathbb{E}_{\tau \sim \mathcal{D}} \big[ \sum_{i=1}^{H} - \log \pi_\theta(a_i | s_i, g(\tau_i), \bar{\tau}_{t-1}^K) \big] \\
& + \alpha \mathbb{E}_{\tau \sim \mathcal{D}_e} \sum_{i=1}^{H} \mathrm{KL} \big[ \pi_\theta(\cdot | s_i, g(\tau_i), \bar{\tau}_{t-1}^K) \| \pi^e(\cdot | s_i) \big] \\
& - \frac{\bar{\eta}}{\mathbb{E}_{\tau \sim \mathcal{D}} \mathbb{E}_{(s_i, a_i) \sim \tau} | Q_\psi(s_i, a_i)|} \mathbb{E}_{\tau \sim \mathcal{D}} \mathbb{E}_{s_i \sim \tau, a_i \sim \pi_\theta} [Q_\psi(s_i, a_i)].
\end{aligned}
\tag{37}
$$

(37) involves four key hyperparameters that may vary across different tasks:

- The weight of the NLL, $\kappa$;
- The weight of the expert policy KL regularizer, $\alpha$;
- The weight of the Q-value guidance term, $\bar{\eta}$;
- The proportion of expert-level trajectories selected from the dataset $\mathcal{D}$ to form the expert dataset $\mathcal{D}_e$, denoted as $\rho\%$.

We employ the *Tree-structured Parzen Estimator (TPE)* for hyperparameter optimization in each environment. The optimal values are determined based on performance on a held-out validation set, aiming to achieve the best trade-off between policy regularization and value maximization. The key hyperparameters of RVDT is given in Table 10. Other hyperparameter configurations used for reproducing the experiments are provided in the code available in the supplementary material.

Table 10: Key hyperparameter settings for all selected tasks to reproduce the main results reported in Table 1.

| Tasks | $\alpha$ | $\bar{\eta}$ | $\kappa$ | Expert dataset ratio $\rho\%$ | Grad Norm |
|---|---|---|---|---|---|
| halfcheetah-medium-expert-v2 | 0.1 | 5.0 | 0.02 | 50% | 15.0 |
| hopper-medium-expert-v2 | 0.01 | 1.0 | 0.02 | 50% | 9.0 |
| walker2d-medium-expert-v2 | 0.01 | 2.0 | 5e-6 | 10% | 5.0 |
| halfcheetah-medium-v2 | 0.1 | 5.0 | 0.02 | 10% | 15.0 |
| hopper-medium-v2 | 1.0 | 1.0 | 0.02 | 50% | 9.0 |
| walker2d-medium-v2 | 0.1 | 2.0 | 0.02 | 10% | 5.0 |
| halfcheetah-medium-replay-v2 | 1.0 | 5.0 | 2e-4 | 50% | 15.0 |
| hopper-medium-replay-v2 | 1.0 | 3.0 | 0.02 | 50% | 9.0 |
| walker2d-medium-replay-v2 | 0.1 | 2.0 | 0.02 | 10% | 5.0 |
| maze2d-open-v0 | 1.0 | 0.01 | 0.02 | 10% | 9.0 |
| maze2d-umaze-v1 | 1.0 | 5.0 | 1.0 | 30% | 20.0 |
| maze2d-medium-v1 | 0.1 | 5.0 | 5e-6 | 30% | 9.0 |
| maze2d-large-v1 | 1.0 | 4.0 | 2e-6 | 30% | 9.0 |
| antmaze-umaze-v0 | 0.1 | 0.05 | 0.02 | 30% | 9.0 |
| antmaze-umaze-diverse-v0 | 0.1 | 0.02 | 0.01 | 70% | 9.0 |
| pen-human-v1 | 10.0 | 0.1 | 2e-3 | 50% | 9.0 |
| hammer-human-v1 | 1.0 | 1.0 | 0.02 | 50% | 5.0 |
| pen-cloned-v1 | 10.0 | 0.1 | 0.02 | 30% | 9.0 |
| hammer-cloned-v1 | 0.5 | 0.01 | 2e-4 | 50% | 9.0 |
| kitchen-complete-v0 | 0.1 | 0.001 | 1e-4 | 50% | 9.0 |
| kitchen-partial-v0 | 10.0 | 0.01 | 0.02 | 50% | 9.0 |
| maze2d-open-dense-v0 | 1.0 | 0.01 | 0.02 | 10% | 9.0 |
| maze2d-umaze-dense-v1 | 0.5 | 3.0 | 0.02 | 10% | 5.0 |
| maze2d-medium-dense-v1 | 100.0 | 5.0 | 2e-4 | 10% | 9.0 |
| maze2d-large-dense-v1 | 1.0 | 4.0 | 2e-3 | 10% | 9.0 |
| antmaze-medium-diverse-v0 | 0.1 | 0.01 | 0.02 | 30% | 9.0 |
| antmaze-large-diverse-v0 | 0.1 | 0.005 | 0.02 | 30% | 9.0 |

## F    Additional Experimental Results

### F.1    Performance in Low-data Regimes

We provide a detailed description of the four sub-datasets $\mathcal{D}_1 - \mathcal{D}_4$ constructed for the low-data regime experiments. First, note that each of the original `maze2d-umaze`, `maze2d-medium`, and `maze2d-large` datasets consists of 2000 trajectories. For each task, we filter the original dataset to distinguish trajectories of different quality levels. We define three quality-based sub-datasets: expert, medium, and noise, denoted as $\mathcal{D}_e$, $\mathcal{D}_m$, and $\mathcal{D}_n$, respectively. These subsets are obtained by randomly sampling 200 trajectories from the top 30%, middle 40%, and bottom 30% of returns in the original dataset.

The sub-datasets $\mathcal{D}_1 - \mathcal{D}_4$ are then constructed by randomly sampling from $\mathcal{D}_e$, $\mathcal{D}_m$, and $\mathcal{D}_n$ according to the following proportions:

1. $\mathcal{D}_1$: 50% $\mathcal{D}_e$ + 50% $\mathcal{D}_m$ + 40% $\mathcal{D}_n$, comprising 280 trajectories, designed to achieve full return-coverage.

2. $\mathcal{D}_2$: 20% $\mathcal{D}_e$ + 20% $\mathcal{D}_m$ + 100% $\mathcal{D}_n$, comprising 280 trajectories, dominated by random-policy data with a smaller fraction of medium and expert trajectories.

3. $\mathcal{D}_3$: 25% $\mathcal{D}_e$ + 25% $\mathcal{D}_m$ + 20% $\mathcal{D}_n$, comprising 140 trajectories, mirroring the empirical return distribution of $\mathcal{D}_1$ but with fewer samples.

4. $\mathcal{D}_4$: 10% $\mathcal{D}_e$ + 10% $\mathcal{D}_m$ + 50% $\mathcal{D}_n$, comprising 140 trajectories, mirroring the empirical return distribution of $\mathcal{D}_2$ with a reduced sample size.

Table 3 demonstrates the significant performance improvements of RVDT over QT in low-data regimes. The percentage improvements of RVDT over QT, derived from Table 3, are summarized in Table 11. These results confirm that RVDT, by incorporating both explicit and implicit rebalancing mechanisms, substantially enhances its ability to recover optimal policies from limited data sam-

ples. This further highlights the non-trivial nature of RVDT's pioneering integration of explicit and implicit rebalancing strategies.

Table 11: Percentage improvement of RVDT over QT across different datasets. For example, an 71.2% ↑ indicates that RVDT outperforms QT by 71.2% on the maze2d-umaze environment when trained with the $\mathcal{D}_1$ dataset.

| **Tasks** (sparse R) | $\mathcal{D}_1$ | $\mathcal{D}_2$ | $\mathcal{D}_3$ | $\mathcal{D}_4$ |
|---|---|---|---|---|
| maze2d-umaze | 71.2% ↑ | 24.4% ↑ | 5.2% ↑ | 63.7% ↑ |
| maze2d-medium | 36.7% ↑ | 8.4% ↑ | 11.5% ↑ | 78.3% ↑ |
| maze2d-large | 28.2% ↑ | 10.6% ↑ | 25.8% ↑ | -5.1% ↓ |
| **Average** | 45.4% ↑ | 14.5% ↑ | 14.2% ↑ | 45.6% ↑ |

## F.2  Impact of Expert dataset ratio

We evaluate the performance of RVDT when trained with varying proportions of subsampled expert trajectories, i.e., by adjusting the expert dataset ratio $\rho\%$. Adjusting the proportion $\rho\%$ directly affects the trade-off between $\alpha_f^*$ and $\alpha_f$: as the proportion increases from 10% to 100%, $\alpha_f^*$ first rises and then declines, while $\alpha_f$ exhibits the opposite trend. This experiment aims to (1) examine how the interplay between $\alpha_f^*$ and $\alpha_f$ influences algorithmic performance, and (2) identify the optimal ratio of expert-level trajectories required for effective learning.

The results, summarized in Table 12, show that using approximately 30% of the top-performing trajectories consistently yields the best results across Maze2D tasks. Specifically, the 30% expert subset achieves higher scores than both smaller (10%) and larger (50%, 70%, and 100%) proportions, suggesting that $\alpha_f^*$ and $\alpha_f$ reach a near-optimal balance under this configuration.

This observation aligns closely with our theoretical analysis, which highlights the importance of maintaining a balance between expert-level and full-spectrum return-coverage. It reveals that effective rebalancing does not simply entail maximizing the proportion of high-quality expert trajectories. Intuitively, relying exclusively on expert data (i.e., maximizing $\alpha_f^*$ while neglecting $\alpha_f$) can cause significant performance degradation due to distributional shift, particularly when training data are limited. Therefore, careful calibration between expert-level and full-spectrum return-coverage is crucial for achieving optimal performance in offline RL.

Table 12: Performance comparison with different proportions of subsampled expert datasets. For example, 10% RVDT corresponds to the performance of RVDT when the expert dataset $\mathcal{D}_e$ consists of the top 10% of trajectories ranked by return from the full offline dataset $\mathcal{D}$.

| **Task** (sparse R) | 10% RVDT | 30% RVDT | 50% RVDT | 70% RVDT | 100% RVDT |
|---|---|---|---|---|---|
| maze2d-umaze-v1 | 93.1 | **143.0** | 100.4 | 102.4 | 101.6 |
| maze2d-medium-v1 | 189.9 | **195.0** | 192.5 | 191.6 | 188.6 |
| maze2d-large-v1 | 246.5 | **250.0** | 248.5 | 248.1 | 243.6 |
| **Average** | 176.50 | **196.0** | 180.47 | 180.70 | 177.93 |

## F.3  Influence of Expert Data Quality

To empirically investigate how the quality of the expert policy affects RVDT performance, we evaluate RVDT under varying proportions of expert trajectory augmentation. As summarized in Table 13, each configuration (e.g., RVDT (15%)) is trained on a mixed dataset composed of the base dataset (merged -m and -m-r) and the top-$k\%$ expert trajectories sampled from the -m-e dataset, which can be regarded as expert-level trajectories.

The results show that RVDT performance consistently improves as the proportion of expert trajectories increases, with the most notable gain observed between RVDT (0%) and RVDT (15%). These findings indicate that higher-quality expert policies contribute to better performance. Combined with the results in Table 12, we can clearly observe the influence of adjusting and rebalancing return

coverage on algorithm performance, thereby further confirming the effectiveness of the proposed return-coverage rebalancing mechanism.

It is important to note that the expert policy used in RVDT for the main experiments (see Table 14) is not trained from any externally provided demonstrations. Instead, it is derived from a subset of the existing offline dataset, denoted as $\mathcal{D}_e \subset \mathcal{D}$, by selecting relatively high-return trajectories. Consequently, our method is fully applicable to standard offline RL settings without requiring any additional expert data. Moreover, the return coverage assumed in our theoretical analysis is not restricted to strictly optimal trajectories. In practice, the presence of near-expert or moderately high-return trajectories is often sufficient to ensure effective learning. This property underscores that RVDT does not rely on idealized expert-level datasets and remains robust as long as the dataset exhibits sufficient diversity to support relative return ranking. As demonstrated in Table 14, RVDT continues to achieve notable improvements even on the -medium and -medium-replay datasets, which contain no expert demonstrations, validating its practicality under realistic offline RL conditions.

Table 13: Performance comparison under varying proportions of expert trajectory augmentation. Each configuration (e.g., RVDT (15%)) is trained on a mixed dataset composed of the base dataset (merged `-m` and `-m-r`) and top-$k\%$ expert trajectories sampled from the `-m-e` dataset.

| Task | RVDT (0%) | RVDT (15%) | RVDT (30%) | RVDT (50%) |
|------|-----------|------------|------------|------------|
| halfcheetah | 49.3 | 92.7 | 93.3 | 94.4 |
| hopper | 100.2 | 113.1 | 113.8 | 113.9 |
| walker2d | 92.7 | 110.3 | 112.4 | 113.8 |
| **Average** | 80.3 | 105.4 | 106.5 | 107.0 |

Table 14: Performance comparison across D4RL tasks. For each task, RVDT reports the mean and standard error of normalized scores [16] over 30 random rollouts and averaged over 5 random seeds.

| Gym Tasks | CQL | IQL | BCQ | TD3+BC | MoRel | BC | DD | DT | StAR | GDT | CGDT | QT | RVDT |
|-----------|-----|-----|-----|--------|-------|----|----|----|------|-----|------|-----|------|
| halfcheetah-m-e | 91.6 | 86.7 | 69.6 | 90.7 | 53.3 | 55.2 | 90.6 | 86.8 | 93.7 | 93.2 | 93.6 | 93.2 ± 0.8 | **94.4 ± 0.1** |
| hopper-m-e | 105.4 | 91.5 | 109.1 | 98.0 | 108.7 | 52.5 | 111.8 | 107.6 | 111.1 | 111.1 | 107.6 | 113.0 ± 0.2 | **113.1 ± 0.5** |
| walker2d-m-e | 108.8 | 109.6 | 67.3 | 110.1 | 95.6 | 107.5 | 108.8 | 108.1 | 109.0 | 107.7 | 109.3 | 112.0 ± 0.3 | **112.7 ± 1.6** |
| halfcheetah-m | 49.2 | 47.4 | 41.5 | 48.4 | 42.1 | 42.6 | 49.1 | 42.6 | 42.9 | 42.9 | 43.0 | 51.0 ± 0.2 | **51.9 ± 0.3** |
| hopper-m | 69.4 | 66.3 | 65.1 | 59.3 | 95.4 | 52.9 | 79.3 | 67.6 | 59.5 | 77.1 | 96.9 | 99.6 ± 1.3 | **100.2 ± 0.1** |
| walker2d-m | 83.0 | 78.3 | 52.0 | 83.7 | 77.8 | 75.3 | 82.5 | 74.0 | 73.8 | 76.5 | 79.1 | 87.2 ± 1.0 | **90.2 ± 0.1** |
| halfcheetah-m-r | 45.5 | 44.2 | 34.8 | 44.6 | 40.2 | 36.6 | 39.3 | 36.6 | 36.8 | 40.5 | 40.4 | 48.8 ± 1.5 | **53.8 ± 2.0** |
| hopper-m-r | 95.0 | 94.7 | 31.1 | 60.9 | 93.6 | 18.1 | 100.0 | 82.7 | 29.2 | 85.3 | 93.4 | 102.1 ± 0.6 | **103.2 ± 1.9** |
| walker2d-m-r | 77.2 | 73.9 | 13.7 | 81.8 | 49.8 | 32.3 | 75.0 | 79.4 | 39.8 | 77.5 | 78.1 | 97.8 ± 0.9 | **99.3 ± 0.8** |
| Average | 80.6 | 77.0 | 53.8 | 75.3 | 72.9 | 52.6 | 81.8 | 76.2 | 66.2 | 79.1 | 82.4 | 89.4 | **91.2** |

| Adroit Tasks | CQL | IQL | BCQ | BEAR | O-RL | BC | DD | D-QL | DT | StAR | GDT | QT | RVDT |
|--------------|-----|-----|-----|------|------|----|----|------|----|------|-----|-----|------|
| pen-human | 37.5 | 71.5 | 66.9 | -1.0 | 90.7 | 63.9 | 66.7 | 72.8 | 79.5 | 77.9 | 92.5 | 111.9 ± 11.8 | **127.2 ± 5.5** |
| hammer-human | 4.4 | 1.4 | 0.9 | 0.3 | 0.2 | 1.2 | 1.9 | 0.2 | 3.7 | 3.7 | 5.5 | 10.4 ± 5.3 | **24.0 ± 1.5** |
| pen-cloned | 39.2 | 37.3 | 50.9 | 26.5 | 60.0 | 37.0 | 42.8 | 57.3 | 75.8 | 33.1 | 86.2 | 85.8 ± 12.6 | **117.8 ± 8.6** |
| hammer-cloned | 2.1 | 2.1 | 0.4 | 0.3 | 2.0 | 0.6 | 1.7 | 3.1 | 3.0 | 0.3 | 8.9 | 11.8 ± 5.8 | **21.3 ± 2.7** |
| Average | 20.8 | 28.1 | 29.8 | 6.5 | 38.2 | 25.7 | 28.3 | 33.4 | 40.5 | 28.8 | 48.3 | 55.0 | **72.6** |

| Kitchen Tasks | CQL | IQL | BCQ | BEAR | O-RL | BC | DD | D-QL | DT | StAR | GDT | QT | RVDT |
|---------------|-----|-----|-----|------|------|----|----|------|----|------|-----|-----|------|
| kitchen-Comp. | 43.8 | 62.5 | 8.1 | 0.0 | 2.0 | 65.0 | 65.0 | 84.0 | 50.8 | 40.8 | 43.8 | 81.7 ± 1.7 | **84.5 ± 2.3** |
| kitchen-partial | 49.8 | 46.3 | 18.9 | 13.1 | 35.5 | 33.8 | 57.0 | 60.5 | 57.9 | 12.3 | 73.3 | 72.5 ± 2.5 | **75.0 ± 2.5** |
| Average | 46.8 | 54.4 | 13.5 | 6.6 | 18.8 | 49.4 | 61.0 | 72.2 | 54.4 | 26.6 | 58.6 | 77.1 | **79.8** |

| Maze2D Tasks | CQL | IQL | BCQ | BEAR | TD3+BC | BC | Diffuser | DD | DT | GDT | QDT | QT | RVDT |
|--------------|-----|-----|-----|------|--------|----|----------|----|----|-----|-----|-----|------|
| maze2d-u | 94.7 | 42.1 | 49.1 | 65.7 | 14.8 | 88.9 | 113.9 | 116.2 | 31.0 | 50.4 | 57.3 | 99.2 ± 1.0 | **145.1 ± 3.8** |
| maze2d-m | 41.8 | 34.9 | 17.1 | 25.0 | 62.1 | 38.3 | 121.5 | 122.3 | 8.2 | 7.8 | 13.3 | 168.8 ± 8.5 | **183.5 ± 4.5** |
| maze2d-l | 49.6 | 61.7 | 30.8 | 81.0 | 88.6 | 1.5 | 123.0 | 125.9 | 2.3 | 0.7 | 31.0 | 242.7 ± 7.3 | **254.3 ± 4.6** |
| Average | 62.0 | 46.2 | 32.3 | 57.2 | 55.2 | 42.9 | 119.5 | 121.5 | 13.8 | 19.6 | 33.9 | 170.2 | **194.3** |

| AntMaze Tasks | CQL | IQL | BCQ | BEAR | TD3+BC | BC | DD | D-QL | DT | StAR | GDT | QT | RVDT |
|---------------|-----|-----|-----|------|--------|----|----|------|----|------|-----|-----|------|
| antmaze-u | 74.0 | 87.5 | 78.9 | 73.0 | 78.6 | 54.6 | 73.1 | 93.4 | 59.2 | 51.3 | 76.0 | 96.0 ± 3.0 | **98.0 ± 4.0** |
| antmaze-u-d | 84.0 | 62.2 | 55.0 | 61.0 | 71.4 | 45.6 | 49.2 | 66.2 | 53.0 | 45.6 | 69.0 | 92.0 ± 6.2 | **98.0 ± 4.0** |
| antmaze-m-d | 53.7 | 70.0 | 0.0 | 8.0 | 3.0 | 0.0 | 24.6 | **78.6** | 0.0 | 0.0 | 6.0 | 24.0 ± 5.6 | 30.0 ± 6.3 |
| antmaze-l-d | 14.9 | 47.5 | 2.2 | 0.0 | 0.0 | 0.0 | 7.5 | **56.6** | 0.0 | 0.0 | 0.0 | 10.0 ± 0.0 | 10.0 ± 0.0 |
| Average | 56.6 | 66.8 | 34.0 | 35.5 | 38.2 | 25.0 | 38.6 | **73.7** | 28.0 | 24.2 | 37.8 | 57 | 59.0 |

## F.4 Details for Ablation Studies

The ablation variants focus on three key aspects: explicit rebalancing mechanisms, implicit rebalancing mechanisms, and policy types. Explicit rebalancing mechanisms include expert-policy KL regularization and expert-level data duplication. For the latter, the duplicated expert data correspond to the top 50% of trajectories ranked by return, with a single additional copy included. The implicit rebalancing mechanism refers specifically to Q-value guidance. Regarding policy types, we consider two policy types, Gaussian stochastic policies and deterministic policies.

Table 15: Component-level breakdown across ablation variants.

| Component | DT | DT-Dup | RDT | QT | VDT | RVDT-Dup | RVDT-Determ | **RVDT** |
|---|---|---|---|---|---|---|---|---|
| Explicit Rebal. | None | Dup. | KL | None | None | Dup. | KL | KL |
| Implicit Rebal. | None | None | None | Q-value | Q-value | Q-value | Q-value | Q-value |
| Policy Type | Determ. | Stoch. | Stoch. | Determ. | Stoch. | Stoch. | Determ. | Stoch. |

Detailed descriptions of the ablation variants in Table 15 are as follows:

- **DT**: The original Decision Transformer method utilizing deterministic policies, implemented by disabling the Q-value guidance term of QT.

- **DT-Dup**: A stochastic variant of DT enhanced with expert data duplication, modeling the policy as a parameterized Gaussian distribution.

- **RDT**: An RVDT variant without the implicit rebalancing mechanism (Q-value guidance), isolating the effect of the expert-policy KL regularizer.

- **QT**: The original QT model implemented using the official codebase, serving as the deterministic-policy counterpart of RVDT without explicit rebalancing mechanisms.

- **VDT**: An RVDT variant excluding expert-policy KL regularization to specifically isolate the contribution of Q-value guidance.

- **RVDT-Dup**: An RVDT variant employing expert-level data duplication as the explicit rebalancing mechanism, enabling comparison between KL-based rebalancing and data duplication approaches.

- **RVDT-Determ**: A deterministic policy variant of RVDT, replacing the stochastic policy with deterministic outputs and substituting KL regularization with an MSE loss, assessing the role of policy expressiveness.

- **RVDT**: The complete RVDT method integrating expert-policy KL regularization and Q-value guidance with a stochastic policy parameterized by a Gaussian distribution.

## F.5 Influence of Policy Expressiveness on Performance

To further investigate the influence of policy expressiveness [4] on algorithmic performance, we compare RVDT with several representative Q-learning-based baselines (CQL, IQL, and D-QL) on the D4RL benchmark. As shown in Table 16, CQL and IQL employ Gaussian stochastic policies, while D-QL adopts a more expressive diffusion policy. In contrast, RVDT is built upon conditional sequence modeling (CSM) techniques and incorporates a simple Gaussian stochastic policy.

From the results, we observe that D-QL, which leverages the diffusion policy, significantly outperforms the simpler Gaussian-policy baselines (CQL and IQL), both across most tasks and in terms of overall average performance, demonstrating the advantage of higher policy expressiveness. Nevertheless, RVDT achieves superior performance on most tasks, which highlights the effectiveness of leveraging CSM together with the return-coverage rebalancing mechanism.

However, in the AntMaze tasks with larger mazes (-m, -l), D-QL performs better than RVDT. This may be attributed to the multi-legged morphology of the Ant agent, which entails a complex and highly multi-modal action distribution. In these scenarios, the expressiveness of the diffusion policy becomes particularly beneficial, whereas RVDT's Gaussian policy may be insufficient to capture such rich multimodal behaviors.

Table 16: Performance of Q-learning-based baselines (CQL, IQL, D-QL) and RVDT on D4RL tasks, where CQL and IQL adopt Gaussian policies, and D-QL employs a diffusion policy. Results are averaged over 5 random seeds.

| Tasks | CQL | IQL | D-QL | RVDT |
|---|---|---|---|---|
| pen-human | 37.5 | 71.5 | 72.8 | **127.2** |
| hammer-human | 4.4 | 1.4 | 0.2 | **24.0** |
| pen-cloned | 39.2 | 37.3 | 57.3 | **117.8** |
| hammer-cloned | 2.1 | 2.1 | 3.1 | **21.3** |
| kitchen-Comp. | 43.8 | 62.5 | 84.0 | **84.5** |
| kitchen-partial | 49.8 | 46.3 | 60.5 | **75.0** |
| maze2d-m | 41.8 | 34.9 | 91.0 | **183.5** |
| maze2d-l | 49.6 | 61.7 | 200.7 | **254.3** |
| antmaze-u | 74.0 | 87.5 | 93.4 | **98.0** |
| antmaze-u-d | 84.0 | 62.2 | 66.2 | **98.0** |
| antmaze-m-d | 53.7 | 70.0 | **78.6** | 30.0 |
| antmaze-l-d | 14.9 | 47.5 | **56.6** | 10.0 |
| Average | 41.2 | 48.7 | 72.0 | **93.5** |

Overall, these findings suggest that both policy expressiveness and CSM with return-coverage rebalancing are effective factors in improving offline RL performance. The two mechanisms are complementary: diffusion policies enhance local action diversity, while CSM with rebalancing strengthens long-horizon modeling capability. Exploring their integration represents a promising direction for future research.

## F.6 Training Time

To evaluate the computational efficiency of RVDT, we provide a detailed comparison of training times among DT, QT, and RVDT on the D4RL benchmark. The results, summarized in Table 17, report the total training time measured until convergence (120 epochs) on an NVIDIA GeForce RTX 4090 GPU. The reported time for RVDT additionally includes the training phase of the expert policy $\pi_e$. As shown in the table, the expert policy training phase typically accounts for approximately one-fifth of the total RVDT training time. When this phase is excluded, the additional overhead introduced by RVDT relative to QT is less than 10%. It can also be observed that QT itself incurs a substantial increase in training time compared to DT, which primarily stems from its additional Q-value learning process.

Table 17: Training time comparison measured until convergence. The reported training time for RVDT additionally includes the expert policy training phase.

| Gym Tasks | DT | QT | RVDT (total) | Expert policy in RVDT |
|---|---|---|---|---|
| halfcheetah-m-e | 7.3hr | 12.2hr | 15.5hr | 3.3hr |
| hopper-m-e | 3.6hr | 10.3hr | 14.6hr | 3.0hr |
| walker2d-m-e | 6.1hr | 11.7hr | 15.4hr | 3.3hr |
| halfcheetah-m | 6.3hr | 11.8hr | 16.1hr | 3.2hr |
| hopper-m | 3.5hr | 9.2hr | 14.0hr | 3.2hr |
| walker2d-m | 6.1hr | 11.5hr | 16.0hr | 3.2hr |
| halfcheetah-m-r | 7.2hr | 12.2hr | 17.4hr | 3.3hr |
| hopper-m-r | 3.5hr | 8.5hr | 12.3hr | 3.2hr |
| walker2d-m-r | 6.2hr | 11.2hr | 14.8hr | 2.8hr |
| **Average** | 5.5hr | 11.0hr | 15.2hr | 3.2hr |

# G    Discussion

## G.1    Imbalance of Datasets

Learning from imbalanced datasets has long been recognized as a critical challenge in offline re-inforcement learning. Such imbalance typically manifests in datasets predominantly composed of low-return trajectories, with relatively few high-return trajectories. This scenario is common in real-world applications since acquiring high-return trajectories is usually more costly compared to low-return ones.

Existing offline RL algorithms, including dynamic programming (DP) based methods, imitation learning (IL) based methods, and conditional sequence modeling (CSM) based methods, encounter significant difficulties when learning from imbalanced datasets. DP-based methods often incorporate inductive biases or regularization terms to enforce policy closeness to the behavioral distribution observed in the dataset, thereby mitigating out-of-distribution (OOD) errors [17, 9], which in turn leads the learned policy to resemble the action distribution of the non-optimal behavior policy. IL-based methods essentially imitate the underlying suboptimal behavior policy that generated the imbalanced dataset, which can result in overly conservative policies due to excessive imitation of actions from the abundant low-performing trajectories. CSM-based methods typically condition on optimal (or near-optimal) returns during inference for decision-making. However, due to the skewed distribution of near-optimal versus low-return trajectories in the training datasets, CSM-based methods frequently train on trajectories with low returns. This discrepancy in return conditioning between the training and inference phases leads CSM methods to experience severe distributional shift problems when conditioning on high returns for inference.

Recent studies describe dataset imbalance from multiple perspectives. For instance, Hong et al. [23] define dataset imbalance through the positive-sided variance of returns (RPSV, defined in Definition 1). Other research interprets dataset imbalance from the perspective of data coverage, where data coverage broadly encompasses both the coverage of state and action spaces [68] and the trajectory return-coverage within the return space [6].

**Definition 1.** *RPSV of a dataset, $\mathbb{V}_+[g(\tau_i)]$, corresponds to the second-order moment of the positive component of the difference between trajectory return: $g(\tau_i) := \sum_{t=0}^{T_i-1} \gamma^t r(s_t^i, a_t^i)$ and its expectation, where $\tau_i$ denotes a trajectory in the dataset:*

$$\mathbb{V}_+[g(\tau_i)] \doteq \mathbb{E}_{\tau_i \sim \mathcal{D}} \left[ (g(\tau_i) - \mathbb{E}_{\tau_i \sim \mathcal{D}}[g(\tau_i)])_+^2 \right], with \quad x_+ = \max\{x, 0\}.$$

## G.2    Expert-Level and Full-Spectrum Return-Coverage

In this paper, we uncover the relationship between the return distribution of the training dataset and the inference-time performance of CSM policies. Since the return distribution of an offline RL dataset corresponds to a discrete distribution over the return space, their probability density functions (PDFs) or cumulative distribution functions (CDFs) are challenging to represent. Therefore, this paper interprets return distributions from the perspective of return-coverage, offering a mathematically tractable description suitable for our theoretical analysis.

It is commonly understood that for a well-trained CSM algorithm, its inference-time performance largely aligns with the initial input return-to-go (RTG) token, which is a manually specified mapping from the initial state $s_1$ to a real number, i.e., $f(s_1) : \mathcal{S} \to \mathbb{R}$. The initial RTG token represents the human-expected final return for the test algorithm; thus, it is generally chosen to correspond to the optimal return associated with the optimal policy $\pi^*$ of the underlying MDP. We denote this optimal policy-derived conditioning function as the *optimal conditioning function $f^*$*. The inference-time returns associated with $f^*$ are referred to as the *optimal conditional returns*. For CSM algorithms, apart from the initial RTG token derived directly from $f^*$, subsequent RTG tokens are computed by subtracting the sampled reward $r$ from the previous RTG token, i.e., $f(s_{t+1}) = f(s_t) - r$. Due to the inherent randomness of $r$ and the stochastic nature of environmental transitions, subsequent RTG tokens cannot consistently remain optimal [5, 13]. Consequently, returns encountered during policy execution follow a more general conditioning function $f$, which with high probability differs from $f^*$. We term this broader class of conditioning functions as *runtime conditioning functions*, and the corresponding inference-time returns as *runtime conditional returns*.

Based on the preceding descriptions of optimal conditional returns and runtime conditional returns, we define their corresponding coverage as *expert-level return-coverage* and *full-spectrum return-coverage*, respectively (consistent with definitions presented in Section 4.1). Let $P_{\pi_\beta}$ denote the distribution over states, actions, and returns induced by the behavior policy $\pi_\beta$. We define the *expert-level return-coverage* in an offline dataset generated by behavior policy $\pi_\beta$ as $P_{\pi_\beta}(g = f^*(s_1)|s_1)$, where $f^*$ corresponds to the optimal conditioning function. Similarly, the *full-spectrum return-coverage* is defined as $P_{\pi_\beta}(g = f(s_1)|s_1)$, where $f$ represents the runtime conditioning function.

Based on these definitions, it becomes evident that return-coverage captures the representation of inference-time conditional returns within the dataset. The central theoretical analysis of this paper (detailed in Section 4) aims to explain how expert-level return-coverage and full-spectrum return-coverage collectively influence the performance of CSM policies. Motivated by this theoretical insight, we propose a simple yet effective return-coverage rebalancing mechanism, designed to substantially enhance the performance of state-of-the-art CSM algorithms.

## G.3    Limitations and Future Directions

Several aspects merit further exploration. First, RVDT involves training an additional imitation policy for the return-coverage rebalancing process, which introduces a modest computational overhead. This cost, however, remains manageable and is well justified by the resulting improvements in policy quality. Future work may explore more efficient architectures or parameter-sharing strategies to further reduce this overhead. Second, the current design of RVDT focuses on the purely offline RL setting. Extending RVDT to online or hybrid settings, where limited online interactions are available, could enable the model to dynamically expand return support beyond the static dataset. In addition, integrating RVDT with data augmentation or generative modeling techniques, such as diffusion-based trajectory stitching or sub-trajectory generation [42, 34], may further enhance return support and improve algorithmic robustness. Third, the current rebalancing mechanism relies on trajectory returns as a measure of trajectory quality. In certain real-world domains, such as healthcare or autonomous driving, reliable return estimation can be challenging. In such cases, alternative evaluation criteria or task-specific metrics could be incorporated to approximate trajectory quality, thereby broadening the applicability of the proposed framework.

