# OpenReview forum: "Rebalancing Return Coverage for Conditional Sequence Modeling in Offline Reinforcement Learning"
_NeurIPS.cc/2025/Conference — NeurIPS 2025 poster_

### Official Review · Reviewer_Mnay · 2025-06-25

**Clarity:** 4
**Significance:** 4
**Originality:** 2
**Rating:** 5
**Confidence:** 3

**Summary:**

One can roughly categorize offline RL algorithms as either "reweighted imitation learning" or "Q learning constrained to in-distribution data". The authors focus on the former, in a sub-area called Conditional Sequence Modeling (CSM). In CSM, we condition the trajectory on a desired Reward to Go (RTG) $g$, denoting the remaining reward in a trajectory from time $t$: $\tau_t$

$$ argmin_\pi - \log \pi(a_t | s_t, g(\tau_t), \tau_{t-1}).  $$

It is difficult to predict $g$. Instead, one can learn what is effectively a value function $f_\pi(s) = V_\pi(s)$


$$ argmin_\pi - \log \pi(a_t | s_t, f_\pi(s_t), \tau_{t-1}).  $$

Thus, the problem reduced to finding the actions that lead to a specific return, where the (undiscounted) return is specified by $f$. By setting $f = V_*$, we can recover an optimal policy.

First, the authors analyze the performance gap caused by non-optimal $f$, (i.e., $f \neq V_*$). Then, they provide sample complexity bounds for CSM. Next, they provide the first algorithmic contribution of the paper, termed RDT.

In RDT, the authors select an expert subset of trajectories and learn a policy on this subset using imitation learning. More formally, they learn a policy that learns a return-conditioned policy on the full dataset, while also imitating the expert-subset policy. In other words, the authors can leverage the full dataset for training, while biasing the policy towards "better" trajectories. The authors utilize RDT to propose RVDT, the main contribution of the paper.

RVDT takes the RDT objective and adds a term that represents the Q function. The full RVDT objective contains three terms -- two from RDT and a new Q value term:
- The return-to-Go objective
- The imitation learning objective, for the expert subset of trajectories
- Maximization of $Q_\pi$

The authors test their algorithm across standard offline RL benchmarks, demonstrating that their approach is state of the art. They perform ablations demonstrating that their method scales well with increasingly-expert data, as well as in low-data regimes.

**Questions:**

- Contributions: Each contribution is a bit long and difficult to parse without reading the preliminary section. Could you simplify/summarize these a bit?
- Section 3.2: Can you please add a sentence with a more intuitive explanation of the conditioning function $f$? The $f$ we aim to find is often the RTG of the optimal policy $\pi_*$, correct? $f(s) = \max_\pi \mathbb{E}_{\pi}[g(s)]$?

**Ethical Concerns:**

["NO or VERY MINOR ethics concerns only"]

**Final Justification:**

The authors resolved my main concern about the originality of their approach by pointing out the theoretical contributions. I would improve my originality score to a 3 or 4, but this does not seem possible right now.

**Limitations:**

I do not think the authors list any limitations. I would suggest they consider adding a few sentences about this. One limitation is that their method requires learning an extra imitation learning policy, which can be costly. Especially in scenarios where the performance gap betwee QT and RVDT is negligible.

**Quality:**

4

**Strengths And Weaknesses:**

### Strengths
- The paper is very well written and I enjoyed reading it. It tackles many complex topics, yet the writing style made it easy to follow along.
- Theorem 2 and 3 provide good theoretical as to why the authors' method should succeed
- The RVDT objective seems fairly simple to implement, as it is just the combination of a few common objectives
- RVDT performs very well compared with prior work
- The authors provide sufficient ablations and analysis of their algorithm

### Weaknesses
- My biggest concern is that the author's method is very similar to QT. My understanding is that their objective just has one more term, corresponding to the imitation learning objective from an expert subset. Many experiments in Table 1 demonstrate that the performance of QT is only marginally less than RVDT.
- Tuning hyperparameters $\eta, \alpha$ seems like it could be difficult. The automatic $\eta$ normalization scheme still likely requires hyperparameter tuning based on the variance of Q values.

---

> ### Author Rebuttal · Authors · 2025-07-31
>
> ## Response to Reviewer [Mnay]
>
> We thank the reviewer for the detailed feedback and thoughtful comments. Below, we address each identified concern point-by-point.
>
> ---
>
> ### **Weakness 1: RVDT is similar to QT**
>
> > My biggest concern is that the author's method is very similar to QT. My understanding is that their objective just has one more term, corresponding to the imitation learning objective from an expert subset. Many experiments in Table 1 demonstrate that the performance of QT is only marginally less than RVDT.
>
> **Response:**
>
> We sincerely thank the reviewer for the detailed feedback. The primary contribution of our work lies in the theoretical investigation of how the performance of CSM-based policies is influenced by the return-coverage of both expert-level and full-spectrum conditional returns in offline reinforcement learning. Based on our theoretical analysis, we propose a simple yet effective rebalancing mechanism. Actually, our theoretical analysis also provides a foundation for other approaches, such as prioritized experience replay, importance reweighting, and advantage-weighted sampling. The expert KL regularization term is introduced as a lightweight, plug-in component that can be seamlessly integrated into existing CSM-based algorithms to enhance robustness and performance, including QT. Moreover, Proposition 2 also shows that the performance upper bound of the optimal policy induced by RVDT’s objective is generally higher than that of QT.
>
> Empirical results confirm the efficacy of the simple return-coverage rebalancing mechanism: the added KL regularization consistently improves RVDT upon QT, especially in challenging settings where Q-value estimates are unreliable. As provided in Table 1 of our main paper, RVDT outperforms QT by approximately 20% on Maze2D tasks with sparse rewards. In addition, Section 5.2 (Lines 301–329) presents three targeted experiments designed to demonstrate RVDT’s superiority over QT in terms of: (1) effective utilization of limited expert data, (2) performance under low-data regimes, and (3) sparse reward environment. The corresponding results are reported in Tables 2, 3, and 4. We reproduce Table 3 below as an illustrative example to highlight the superiority of RVDT in low-data regimes.
>
> Table 3: Performance comparison in low-data regimes.
> |Task|QT($D_1$)|RVDT($D_1$)|QT($D_2$)|RVDT($D_2$)|QT($D_3$)|RVDT($D_3$)|QT($D_4$)|RVDT($D_4$)|
> |-|-|-|-|-|-|-|-|-|
> |maze2d-umaze|100.3|171.7|81.8|101.8|73.1|76.9|61.4|100.5|
> |maze2d-medium|137.1|187.4|175.2|190.0|163.2|182.0|98.3|175.3|
> |maze2d-large|109.5|140.4|81.5|90.1|104.4|131.3|100.2|95.1|
> |Average|115.6|166.5|112.8|127.3|113.6|130.1|86.6|123.6|
>
> ---
>
> ### **Weakness 2: Difficulty of Hyperparameter Tuning**
>
> > Tuning hyperparameters $\eta, \alpha$ seems like it could be difficult. The automatic $\eta$ normalization scheme still likely requires hyperparameter tuning based on the variance of Q values.
>
> **Response:**
>
> We appreciate the reviewer’s insightful comment. It is true that RVDT introduces two additional regularization terms on top of the DT loss, which brings some extra hyperparameter tuning effort to ensure optimal performance. We summarize the loss in Equation (7) into the following form:
> $$L(RVDT) = L(DT) - \eta L(Q) + \alpha L(KL).$$
>
> In practice, we find that certain tuning heuristics can effectively reduce the tuning burden, especially when considering the relative reliability of the two regularizers across different types of datasets. For instance, in environments where Q-value estimation is unreliable—such as those with sparse rewards—we typically set $\eta$ to a relatively small value. This shifts the optimization emphasis toward the expert KL regularizer, which tends to provide a more stable and effective learning signal in such scenarios. Conversely, in environments with dense and reliable reward signals, a relatively larger $\eta$ may be preferred to better leverage the Q-regularizer.
> As a result, tuning $\eta$ and $\alpha$ is generally not prohibitively difficult.
>
> From a theoretical perspective, Proposition 1 in our paper provides partial justification for the compatibility of the two regularizers. Specifically, under the Gaussian policy assumption, the expert KL regularizer can be interpreted as implementing a weighted sampling scheme, leading to the following reformulated loss:
> $$ L_{RVDT}(θ) =  E_{τ \sim D} [ (1 + α · I[τ ∈ D_e]) · ( Σ_{i=1}^H -log π_θ(a_i | s_i, g(τ_i), τ̄_{t-1}^K) ) ] - \eta L(Q).$$
> This formulation suggests that the expert KL regularizer effectively operates through a sampling-based mechanism that is inherently compatible with the main DT loss. As such, it does not introduce significant conflict during optimization and maintains stability in practice.
>
> ---
>
> ### **Question 1: Clarity of Contributions**
>
> > Contributions: Each contribution is a bit long and difficult to parse without reading the preliminary section. Could you simplify/summarize these a bit?
>
> **Response:**
>
> We appreciate the reviewer’s request. To improve clarity, we provide a simplified summary of our contributions below.
>
> 1. We theoretically characterize how the performance of CSM-based policies depends on the coverage of high-quality (expert-level) and runtime (full-spectrum) returns in the offline dataset.
> 2. We propose a simple yet effective return-coverage rebalancing mechanism, which can serve as a plug-in module to improve existing CSM-based algorithms.
> 3. We design RVDT, a new CSM-based offline RL algorithm built upon DT, incorporating both Q-value and expert policy KL-divergence regularization for better alignment between the expected returns of sampled actions and the optimal returns.
>
> The current presentation of our contributions in the paper prioritizes completeness, which is why we include relevant technical details to ensure precision and clarity in context.
>
> ---
>
> ### **Question 2: Intuition Behind the Conditioning Function in Section 3.2**
>
> > Section 3.2: Can you please add a sentence with a more intuitive explanation of the conditioning function $f$? The $f$ we aim to find is often the RTG of the optimal policy $\pi^\star$, correct? $f(s) = \max_{\pi}\mathbb{E}_{\pi}[g(s)]$
>
> **Response:**
>
> Thank you for the insightful comment, and your understanding is indeed correct. We will add a more intuitive explanation of the conditioning function $f$ to Section 4.1, immediately following the sentence: "Our analysis extends this perspective by considering a more general function class that encompasses both near-optimal and potentially non-optimal runtime conditioning functions, denoted as $f^\star$ and $f$, respectively." (Line 162). The added explanation will be as follows:
>
> The runtime conditional return function $f$ corresponds to arbitrary possible returns that may be encountered during policy execution. Let $\mathcal{G}$ denote the collection of all possible returns collected by $\pi \in \Pi$, then $f: \mathcal{S} \rightarrow \mathcal{G}$. The target conditional function $f^\star$ that we aim to find is the RTG under the optimal policy $\pi^\star$, i.e., $f(s) = \max_{\pi}\mathbb{E}_{\pi}[g(s)]$.
>
> ---
>
> ### **Limitations: Limitations Not Discussed**
>
> > Limitations: I do not think the authors list any limitations. I would suggest they consider adding a few sentences about this.
>
> **Response:**
>
> We sincerely thank the reviewer for the helpful suggestion. We will add a discussion in our revision to state the limitations of the proposed RVDT algorithm. The content is as follows:
>
> The proposed RVDT algorithm has several limitations:
> 1. RVDT requires learning an additional imitation policy, which introduces extra computational overhead.
> 2. RVDT is currently designed as a purely offline RL algorithm. How to effectively incorporate online interactions to expand return support remains an open question and will be a promising direction for future work.
>
> ---
>
> We sincerely appreciate the reviewer’s feedback and will incorporate these suggestions to improve clarity and completeness in the revised version, and we hope our response can alleviate some of your concerns.

---

> ### Comment · Reviewer_Mnay · 2025-08-01
>
> Thank you for your response. The authors have addressed all my concerns. I agree that the theoretical contributions make up for the similarity between QT and the proposed method. I still consider this paper an accept, and will update the originality score.
>
> EDIT: It does not seem possible to change the originality score in OpenReview. Please consider it as 3 instead of 2.

---

> ### Author Response · Authors · 2025-08-01
>
> Thank you for taking the time to review our paper. We sincerely appreciate your thoughtful feedback and positive recommendation.

---

### Official Review · Reviewer_4BE1 · 2025-06-29

**Clarity:** 3
**Significance:** 3
**Originality:** 3
**Rating:** 4
**Confidence:** 4

**Summary:**

This paper addresses the trade-off between return maximization and behavior coverage in conditional sequence modeling (CSM)-based offline RL methods such as Decision Transformer. They propose a test-time control framework that balances this trade-off by reranking sampled trajectories using a weighted combination of predicted return and a learned coverage score. Their method, which does not alter the training process, shows improved performance across several D4RL tasks.

**Questions:**

1. The combined use of Q-value regularization and expert KL divergence may result in conflicting optimization signals, particularly when Q-value estimation is poor or inaccurate. How do you address or mitigate the potential instability and performance degradation that may arise from this issue?
2. Besides using top-k return for expert trajectory selection, did the authors consider or evaluate any alternative definitions or methods for identifying expertise in the offline dataset?
3. The bound in Theorem 2 shows a dependence on both expert-level and full-spectrum return coverage parameters (α_f and α_f*). Could you provide more intuition or empirical evidence about how trade-offs between these two types of coverage affect policy performance? Are there scenarios where improving one necessarily degrades the other?

**Ethical Concerns:**

["NO or VERY MINOR ethics concerns only"]

**Limitations:**

Yes

**Quality:**

3

**Strengths And Weaknesses:**

Strengths:

1. The paper addresses a meaningful and underexplored problem in conditional sequence modeling (CSM) for offline RL—namely, the trade-off between return maximization and behavioral coverage—which is both practically relevant and conceptually important.
2. The theoretical formulation is relatively complete and provides a useful lens to understand the inherent tension in CSM-based policies between imitating high-return behaviors and maintaining adequate state-action coverage.
3. The experimental design is comprehensive, evaluating across multiple dimensions including data quality (expert vs mixed), data scale (low-data regime), and reward density (sparse vs dense).

Weaknesses:

1. The proposed rebalancing mechanism fundamentally relies on the availability and selection of high-quality, expert-level trajectories. In many real-world offline RL settings, such as medical or autonomous driving applications, “expert” behavior is often ambiguous.
2. While the method claims to enhance expert-level coverage, in practice the KL regularization simply increases the weight of expert-like behaviors, but does not expand the policy’s support into previously unobserved state-action regions.

---

> ### Author Rebuttal · Authors · 2025-07-31
>
> ## Response to Reviewer [4BE1]
>
> We sincerely thank the reviewer for the detailed and insightful comments. We address each question below.
>
> ---
>
> ### **Weakness 1**
>
> > The proposed rebalancing mechanism fundamentally relies on the availability and selection of high-quality, expert-level trajectories...
>
> **Response:**
>
> We sincerely thank the reviewer for raising this important point. Note that the return-coverage required in our analysis is not strictly restricted to optimal trajectories. In practice, the presence of relatively high-quality trajectories or near-expert returns is often sufficient to facilitate meaningful learning outcomes. We would like to clarify that our method does not require any externally provided expert data, and it operates entirely on the given offline dataset and remains effective as long as there is sufficient diversity to enable relative ranking. As demonstrated in Table 1 of the main paper, our method consistently improves performance even on the "-medium" and "-medium-replay" datasets, which contain no expert-level trajectories, thereby highlighting its practicality beyond idealized settings.
>
> Nevertheless, we agree that in certain real-world domains, such as healthcare or autonomous driving, assessing trajectory quality based solely on return may be challenging. In such cases, additional annotations or task-specific metrics may be necessary to approximate trajectory quality, which could then serve as a foundation for applying our rebalancing framework. We will include a discussion of these potential extensions in the revision and consider this an important direction for future work.
>
> ---
>
> ### **Weakness 2**
>
> > The KL regularization simply increases the weight of expert-like behaviors, but does not expand the policy’s support...
>
> **Response:**
>
> We appreciate the reviewer’s insightful observation. Generally speaking, there are two aspects that influence the return-coverage: reward coverage, which is explicitly involved in the definition, and state-action space coverage, which is implicitly involved. To increase return-coverage, we introduce an expert KL regularization that effectively performs weighted sampling over a proportion of high-quality trajectories with high cumulative rewards. As for state-action space coverage, it is generally difficult to expand the policy’s support into previously unobserved state-action regions in offline RL without resorting to data augmentation or generative modeling. Indeed, prior work [1,2] has explored this direction by leveraging model-based approaches or generative models (e.g., diffusion models) to synthesize new transitions via trajectory stitching or sub-trajectory generation. These approaches are discussed in Lines 97–101.
>
> It is important to emphasize that our proposed rebalancing mechanism is orthogonal and complementary to these methods. It can be seamlessly applied on top of augmented or generated datasets to further expand policy support while still leveraging expert KL regularizer. We view the integration of rebalancing mechanisms with advanced data generation techniques as a promising direction for future work.
>
> [1] Guanghe Li, et al. Diffstitch: Boosting offline rl with diffusion-based trajectory stitching. ICML, 2024.
>
> [2] Sungyoon Kim, et al. Stitching sub-trajectories with conditional diffusion model for goal-conditioned offline rl. AAAI, 2024.
>
> ---
>
> ### **Question 1**
>
> > The combined use of Q-value regularization and expert KL divergence may result in conflicting optimization signals...
>
> **Response:**
> We sincerely thank the reviewer for this insightful observation. As described in Eq. (7) of the paper, the loss function of RVDT can be summarized as:
> $$L(RVDT) = L(DT) - \eta L(Q) + \alpha L(KL).$$
>
> **From an empirical perspective**, we mitigate potential instability by carefully balancing the hyperparameters $\eta$ and $\alpha$ associated with the two regularizers. In practice, for datasets where the Q-values are difficult to estimate accurately, such as those with sparse rewards, we typically set $\eta$ to a relatively small value. This shifts the optimization emphasis toward the expert KL regularizer, which serves as a more reliable guiding signal and helps stabilize training in the presence of Q-value estimation errors.
>
> **From a theoretical perspective**, Proposition 1 in our paper provides partial justification for the compatibility of the two regularizers. Specifically, under the Gaussian policy assumption, the expert KL regularizer can be interpreted as implementing a weighted sampling scheme, leading to the following reformulated loss:
> $$ L_{RVDT}(θ) =  E_{τ \sim D} [ (1 + α · I[τ ∈ D_e]) · ( Σ_{i=1}^H -log π_θ(a_i | s_i, g(τ_i), τ̄_{t-1}^K) ) ] - \eta L(Q).$$
> This formulation suggests that the expert KL regularizer effectively operates through a sampling-based mechanism that is inherently compatible with the main DT loss. As such, it does not introduce significant conflict during optimization and maintains stability in practice.
>
> Regarding the concern of **performance degradation**, the dual-regularizer design of RVDT is specifically intended to enhance robustness in challenging settings, such as those with sparse or unreliable reward signals where Q-value estimation is inherently difficult. Our experimental results demonstrate that RVDT consistently outperforms QT in such scenarios, validating the practical effectiveness of this design. Furthermore, as shown in Proposition 2 (lines 241–244), the performance of RVDT is generally higher than that of QT.
>
> In summary, by appropriately tuning $\eta$ and $\alpha$, RVDT effectively mitigates potential conflicts between regularizers and avoids the instability or performance degradation.
>
> ---
>
> ### **Question 2**
>
> > Did the authors consider or evaluate any alternative definitions or methods for identifying expertise in the offline dataset?
>
> **Response:**
>
> We sincerely thank the reviewer for this thoughtful question. Actually, alternative criteria for assessing trajectory quality, such as the averaged Q-value along the trajectory and Conditional Value-at-Risk, can also be adopted. However, these alternatives are more complicated to compute and typically require additional neural network approximation, thus making them less well suited to our current setting and tasks than trajectory return.
>
> Our study primarily focuses on the theoretical analysis of how the performance of conditional sequence modeling (CSM)-based policies is affected by the return-coverage of expert-level and full-spectrum trajectories in the offline training dataset. Building upon our theoretical results, we propose a simple yet effective return-coverage rebalancing mechanism to improve the performance of current CSM-based algorithms, and this mechanism also serves as a plug-in module that can be easily integrated into existing CSM-based algorithms. In this context, using trajectory return as a proxy for expertise is a straightforward criterion within the standard setting of CSM-based algorithms. It aligns well with the return-conditioned training paradigm and provides a natural way to stratify trajectories by quality.
>
> Nevertheless, exploring alternative expert selection criteria, especially in more complex scenarios such as reward-free environments and constrained MDPs, would indeed be meaningful. However, such settings are beyond the scope of the present work. We consider this a promising direction and leave it to future research.
>
> ---
>
> ### **Question 3**
>
> > Could you provide more intuition or empirical evidence about how trade-offs between these two types of coverage affect policy performance?...
>
> **Response:**
>
> We sincerely thank the reviewer for the detailed observation. The performance bound of a CSM-based policy is jointly influenced by $\alpha_f$ and $\alpha_f^{\star}$. In the offline RL setting, the relationship $\alpha_f + \alpha_f^{\star} \leq 1$ holds when $f \neq f^{\star}$, implying that increasing one is likely to result in a decrease in the other. To empirically validate this theoretical insight, we present an experiment in Appendix F.2 (lines 1121-1132), with results summarized in Table 12. This experiment evaluates RVDT’s performance when return-coverage rebalancing is applied using varying proportions of top-$k$ trajectories based on return.
>
> Table 12: Performance comparison with different proportions of subsampled expert datasets.
> |Task (sparse R)|10% RVDT|30% RVDT|50% RVDT|70% RVDT|100% RVDT|
> |-|-|-|-|-|-|
> |maze2d-u-v1|93.1|**143.0**|100.4|102.4|101.6|
> |maze2d-m-v1|189.9|**195.0**|192.5|191.6|188.6|
> |maze2d-l-v1|246.5|**250.0**|248.5|248.1|243.6|
> |Average|176.50|**196.0**|180.47|180.70|177.93|
>
> As shown in the table, varying the proportion of top-$k$ trajectories directly affects the trade-off between $\alpha_f^{\star}$ and $\alpha_f$: as the proportion increases from 10% to 100%, $\alpha_f^{\star}$ first increases and then decreases, while $\alpha_f$ follows the opposite trend. Correspondingly, the performance of RVDT peaks when 30% of the highest-return trajectories are used, suggesting that $\alpha_f$ and $\alpha_f^{\star}$ reach a near-optimal balance under this setting.
>
> This finding supports our theoretical results and emphasizes that effective return-coverage rebalancing is not achieved by simply maximizing the use of expert trajectories. Intuitively, relying exclusively on expert data (i.e., maximizing $\alpha_f^{\star}$ while neglecting $\alpha_f$) can lead to significant performance degradation due to distributional shift, especially when the training data is limited. Therefore, careful calibration and balancing of both expert-level and full-spectrum return-coverage is essential for achieving optimal performance in offline RL.
>
> ---
>
> Once again, we thank the reviewer for their thoughtful feedback, and we hope our response can alleviate some of your concerns and offer useful clarification.

---

> > ### Comment · Reviewer_4BE1 · 2025-08-02
> >
> > Thanks for the response! The authors have answered all my questions. I will keep my score.

---

> ### Author Response · Authors · 2025-08-05
>
> Thank you for taking the time to review our paper. We sincerely appreciate your thoughtful feedback and constructive comments. We are glad that our responses have addressed all your concerns.

---

### Official Review · Reviewer_8NTa · 2025-06-30

**Clarity:** 3
**Significance:** 2
**Originality:** 3
**Rating:** 4
**Confidence:** 4

**Summary:**

This paper explore the CSM used in offline RL. The auther studies two types of return (expert-level and full spectrum) and provide some theoretical analysis on their impact. Additionally, auther provide a return converage rebalancing mechanism into DT framework.

**Questions:**

1. Can auther discuss the validity of the return-coverage assumption? Those assumptions may hold in small or well-curated offline datasets, it becomes increasingly difficult to satisfy as the dataset grows in size and complexity. In such cases, $\alpha$ may become vanishingly small, which would cause the theoretical performance bounds to become vacuous or uninformative.

2. How the quality of the expert policy affects RVDT performance, How long it takes to train the expert policy, and the trade-off between expert policy quality, training time, and RVDT effectiveness?

3. For Table 1 gym tasls, majority of RVDT performance are so close to the QT  if we consider QT's statistic value, it may hard to tell if RVDT has performance different with QT for the gym tasks. Same issue for Kitchen tasks and AntMaze tasks

4. Why is  top 50% of trajectories to construct the expert dataset not more?

I would be willing to raise my rating if the authors can address 1. the assumption bound issue 2. the inconsistance statistic result report, as they are my primary concern and the main reason for my initial assessment.

**Ethical Concerns:**

["NO or VERY MINOR ethics concerns only"]

**Final Justification:**

The reviewer addressed all concerns; however, I cannot give a full acceptance score (5) because relying on an assumption or theorem with clear limitations—regardless of its common use in other works—is not a strong basis for theoretical analysis.

**Limitations:**

yes

**Quality:**

3

**Strengths And Weaknesses:**

>Strength

1. the paper is clear
2. auther did compare with various benchmarks and algorithms

>Weaknesses

1. The major concern is the assumptions in the theorem and the bound value $\alpha_f$. In order for those assumption to hold, the $\alpha$ maybe really small or even close to 0.  If this is the case, the bound for theorem 1 and lemma 1 become meaningless ($\infty$).

2. The method relies on training an expert policy via imitation learning on near-optimal trajectories, which is then used to regularize the main policy through a KL divergence term. The quality of the expert policy will affect the performance of RVDT. Additionally, the higher quality of expert policy will result in longer pretraining time and utilize larger dataset.

3. For the evaluation, some of results have statistic value (std) some are not.

---

> ### Author Rebuttal · Authors · 2025-07-31
>
> ## Response to Reviewer [8NTa]
> We sincerely thank the reviewer for the thoughtful and constructive feedback. We address each point below in detail.
>
> ---
>
> ### **Weakness 1 and Question 1**
> **Response:**
> We provide clarifications to address the reviewer’s concern regarding the return-coverage assumption:
>
> 1. **Alignment with existing literature.**
> The return-coverage assumption we adopt is a standard analytical assumption in most existing CSM-based methods, including works such as DT [1] and QT. In fact, theoretical analyses in most existing offline RL algorithms typically rely on some form of coverage condition, either over return pairs or the state-action space, to ensure that the learning problem is well-posed and admits meaningful generalization guarantees.
> 2. **Interpretation of the theoretical bound.**
> We fully acknowledge the reviewer’s valid concern regarding extremely small return-coverage. In such cases, the theoretical performance gap becomes unbounded, which actually reflects the inherent nature of CSM-based methods. Specifically, when the return-coverage of the training data is insufficient, reliable policy learning cannot be guaranteed, and the performance gap can become arbitrarily large. This phenomenon, as predicted by our theory, has already been widely observed in empirical studies: the more comprehensively a dataset covers expert-level and full-spectrum conditional returns, the better the performance achieved by CSM-based methods.
> 3. **The necessity of coverage in offline RL.**
> Note that the return-coverage required in our analysis is not strictly restricted to optimal trajectories. Indeed, coverage over relatively high-quality or expert-level returns is often sufficient to ensure a meaningful performance gap. Our analysis naturally extends to datasets containing only a subset of high-quality trajectories, which can guide rebalancing in CSM-based methods. The necessity of coverage over relatively high-quality trajectories aligns with common intuition in offline RL. For example, for BC-based methods, generalization depends on sufficient coverage over useful state-action pairs. Similarly, CSM-based methods require non-negligible return-coverage to achieve satisfactory performance. In contrast, if the dataset contains only uniformly poor trajectories, no CSM-based method can be expected to recover a high-quality policy.
> 4. **Practical validity.**
> In practice, as the dataset grows in size, the factors that primarily influence return-coverage are the difficulty of the task and the data collection strategy. Generally, a good data collection strategy is required to obtain sufficient return-coverage for difficult tasks. Actually, widely adopted large-scale offline RL benchmarks (e.g., D4RL) typically include a non-trivial amount of high-quality data, which satisfy the coverage assumption. Practically, high-quality trajectories may be diluted by a large quantity of lower-quality data as the dataset grows. Our rebalancing mechanism plays a crucial role in dynamically adjusting return coverage, thereby keeping the performance gap within a relatively low range.
>
> [1] Brandfonbrener, David, et al. When does RCSL work for offline rl. NIPS. 2022.
>
> ---
>
> ### **Weakness 2 and Question 2**
> **Response:**
> Indeed, the quality of the expert policy influences the performance of RVDT. Nevertheless, we emphasize that the expert policy is trained using a subset $D_e \subset D$ rather than any additional data. It is worth noting that the "expert data" in our method does not necessarily refer to exact expert demonstrations, but rather to relatively high-quality trajectories selected from the existing dataset.
>
> We empirically investigate how expert policy quality impacts RVDT performance. As shown in Table 2, adding more expert trajectories to a fixed non-expert dataset (RVDT(0%)–RVDT(50%)) consistently improves performance. A higher percentage indicates more expert data and thus a higher-quality expert policy used for regularization. See Section 5.2 (lines 301–310) for details.
>
> Table 2: Performance with increasing expert trajectory proportions.
> |Task|RVDT(0%)|RVDT(15%)|RVDT(30%)|RVDT(50%)|
> |-|-|-|-|-|
> |halfcheetah|49.3|92.7|93.3|94.4|
> |hopper|100.2|113.1|113.8|113.9|
> |walker2d|92.7|110.3|112.4|113.8|
> |Average|80.3|105.4|106.5|107.0|
>
> Table 2 shows that RVDT performance consistently improves with better expert policy quality, with the largest gain observed between RVDT(0%) and RVDT(15%). The results reveal the relationship between expert policy quality and RVDT performance, demonstrating both the effectiveness and benefit of our method.
>
> Regarding **training time**, the dominant influencing factor is the number of training epochs, rather than the dataset size. The training time is reported in the table below. The expert policy training time for RVDT was approximately 3 hours on an NVIDIA 4090 GPU, with minimal variation across different tasks.
>
> Table: Training time comparison, where RVDT's training time includes the expert policy training phase.
> |Gym Tasks|DT|QT|RVDT (total)|Expert policy|
> |-|-|-|-|-|
> |halfcheetah-m-e|7.3hr|12.2hr|15.5hr|3.3hr|
> |hopper-m-e|3.6hr|10.3hr|14.6hr|3.0hr|
> |walker2d-m-e|6.1hr|11.7hr|15.4hr|3.3hr|
>
> To address the **trade-off** between expert policy quality, training time, and RVDT effectiveness, we primarily rely on tuning several key hyperparameters within the algorithm.
> Note that our primary goal is to find a policy that achieves the highest possible performance given fixed training data. Therefore, rather than explicitly trading off between the three, we search for optimal hyperparameters. Specifically, RVDT selects the top $\rho$% of high-quality trajectories to train the expert policy until convergence. Then, it adjusts the coefficients $\eta$ and $\alpha$ in the overall loss function: $L(RVDT) = L(DT) - \eta L(Q) + \alpha L(KL)$. In practice, we perform grid search over $\rho$, $\eta$, and $\alpha$, and select the combination that yields the best performance.
>
> ---
>
> ### **Weakness 3 and Question 3**
> **Response:**
> We appreciate the reviewer’s emphasis on statistical rigor. Following common convention in CSM-based offline RL research (e.g., DT, CQL, QT), we report both the mean and standard deviation (std.) for our own method, while only reporting mean values for other baselines, consistent with prior works. Note that some previous works do not report std., and it is generally infeasible for us to report std. for all baselines.
>
> Nevertheless, given the importance of QT, we will include its std. in the revised manuscript. A subset of the results is shown in the table below due to space limitations. The results demonstrate that RVDT consistently outperforms QT, even when accounting for the variance. Notably, QT exhibits higher variance than RVDT, largely due to its sensitivity to Q-value estimation errors. In contrast, the KL regularization term in RVDT helps mitigate the impact of such errors. Furthermore, according to Proposition 2 (lines 241–244), the performance of RVDT would generally be better than that of QT.
>
> |Gym Tasks|QT|RVDT|
> |-|-|-|
> |halfcheetah-m-e|93.2 ± 0.8|94.4 ± 0.1|
> |hopper-m-e|113.0 ± 0.2|113.1 ± 0.5|
> |walker2d-m-e|112.0 ± 0.3|112.7 ± 1.6|
> |...|...|...|
> |pen-human|111.9 ± 11.8|127.2 ± 5.5|
> |hammer-human|10.4 ± 5.3|24.0 ± 1.5|
> |pen-cloned|85.8 ± 12.6|117.8 ± 8.6|
> |hammer-cloned|11.8 ± 5.8|21.3 ± 2.7|
> |...|...|...|
> |antmaze-u-d|92.0 ± 6.2|98.0 ± 4|
> |antmaze-m-d|24.0 ± 8.6|30.0 ± 6.3|
> |antmaze-l-d|10.0 ± 0|10.0 ± 0|
>
> Regarding the reviewer’s concern that RVDT shows only modest gains over QT on certain tasks (e.g., MuJoCo), this is mainly due to boundary effects: both methods already reach or exceed the dataset’s expert-level returns, leaving any further improvement inherently challenging.
>
> RVDT’s advantage over QT is most evident when accurate Q-value estimation is difficult, such as in low-data or sparse-reward settings. As shown in Table 3 (lines 311–320), RVDT significantly outperforms QT under low-data regimes. The four datasets used here contain only about one-tenth of the original D4RL data, with difficulty increasing from $D_1$ to $D_4$ (see Appendix F.1 for details). RVDT consistently outperforms QT, achieving average improvements of 45.4%, 14.5%, 14.2%, and 45.6% over QT on $D_1$–$D_4$, respectively. Additional evidence of RVDT’s advantage in sparse-reward settings is shown in Table 4 (lines 321–329), where it outperforms QT by 27%, 38%, 7%, and 6% on maze2d-open, maze2d-umaze, maze2d-medium, and maze2d-large, respectively. Collectively, these results confirm RVDT’s clear and consistent superiority over QT under challenging conditions.
>
> Table 3: Performance comparison in low-data regimes.
> |Task|QT($D_1$)|RVDT($D_1$)|QT($D_2$)|RVDT($D_2$)|QT($D_3$)|RVDT($D_3$)|QT($D_4$)|RVDT($D_4$)|
> |-|-|-|-|-|-|-|-|-|
> |maze2d-u|100|171|81|101|73|77|61|100|
> |maze2d-m|137|187|175|190|163|182|98|175|
> |maze2d-l|109|140|81|90|104|131|100|95|
> |Average|115|166|112|127|113|130|86|124|
>
> ---
>
> ### **Question 4**
>
> **Response:**
> In fact, the percentage of top $\rho$% trajectories selected to construct the expert dataset is a tunable hyperparameter. For our main results reported in Table 1 (Section 5.1 of the manuscript), the specific values used for this hyperparameter are provided in Appendix E (Table 10), and they vary across tasks. This hyperparameter is selected via grid search.
>
> For ablation studies, unless otherwise specified, we adopt the 50% setting primarily for the purpose of controlled comparison. One reason for choosing 50% is that it represents the most commonly used value across tasks (see Table 10), making it a representative and balanced choice for the general analysis.
>
> ---
>
> Again, we thank the reviewer for their valuable comments and suggestions. We hope our response can alleviate some of your concerns and offer useful clarification.

---

> > ### Comment · Reviewer_8NTa · 2025-08-04
> >
> > Thanks for the detailed response which now I fully aware, the limitation of theoretical analysis is general. I will increase my score.

---

> ### Author Response · Authors · 2025-08-05
>
> Thank you very much for your thoughtful follow-up and for taking the time to re-evaluate our work. We are glad that our clarifications regarding the return-coverage assumption have effectively addressed your concerns. Indeed, this assumption is widely adopted and recognized as standard and reasonable within the conditional sequence modeling literature. We sincerely appreciate your updated evaluation and support.

---

### Official Review · Reviewer_Lvw8 · 2025-07-03

**Clarity:** 3
**Significance:** 3
**Originality:** 3
**Rating:** 4
**Confidence:** 5

**Summary:**

This paper investigates the distributional shift issue in conditional sequence modeling (CSM) for offline RL, particularly caused by imbalanced return distributions. The authors highlight the importance of covering both expert-level and full-spectrum returns, proposing a straightforward return-coverage rebalancing mechanism named Rebalancing Value-regularized Decision Transformer (RVDT). Extensive experiments on D4RL benchmarks validate that RVDT achieves state-of-the-art performance.

**Questions:**

See the weaknesses part above.

**Ethical Concerns:**

["NO or VERY MINOR ethics concerns only"]

**Final Justification:**

Thank you for your detailed responses, which have addressed most of my concerns. However, the additional computational overhead appears to offer limited cost-effectiveness. I will keep my original rating but increase my confidence score to reflect a leaning toward acceptance.

**Limitations:**

Yes.

**Paper Formatting Concerns:**

No formatting concern.

**Quality:**

3

**Strengths And Weaknesses:**

**Strengths:**

1. The findings presented in this paper, particularly regarding the significant impact of expert-level coverage and full-spectrum conditional returns, are important and valuable to the research community.
2. Based on the analysis provided above, the proposed RVDT method is both simple and effective.
3. The experiments conducted are thorough and comprehensive.

**Weaknesses:**

1. Please correct me if I have overlooked anything, but an important ablation study seems to be missing. Specifically, an ablation on the two newly introduced terms in Equation (7)—the policy regularization term and the value-based guidance term—would be helpful. What would be the impact on performance if either of these two terms were removed?
2. In the experimental section, the authors attribute RVDT's inferior performance compared to D-QL on the larger AntMaze tasks to D-QL's highly expressive policies. However, due to the lack of further experimental evidence, it would be helpful if the authors could clarify how D-QL performs relative to RVDT on larger Maze2D tasks, which are similarly large maze-like environments. Alternatively, could the authors provide additional qualitative or quantitative results to support this claim?
3. What is the computational overhead introduced by RVDT? For instance, RVDT requires first training an expert policy \$\pi\_e\$ and then introduces two additional terms during offline RL training.
4. Some related works also utilize a small portion of expert data to improve offline RL performance \[1,2,3].

\[1] Kumar, Aviral, et al. "Pre-training for robots: Offline RL enables learning new tasks from a handful of trials." *arXiv preprint arXiv:2210.05178* (2022).

\[2] Yang, Qisen, et al. "Hundreds guide millions: Adaptive offline reinforcement learning with expert guidance." *IEEE Transactions on Neural Networks and Learning Systems* 35.11 (2023): 16288–16300.

\[3] Cheng, Peng, et al. "Pushing the Limit of Sample-Efficient Offline Reinforcement Learning." *ICLR 2025 Workshop on World Models: Understanding, Modelling and Scaling.*

---

> ### Author Rebuttal · Authors · 2025-07-31
>
> ## Response to Reviewer [Lvw8]
>
> We sincerely thank the reviewer for the constructive and insightful feedback. Below, we address each concern point-by-point.
>
> ---
>
> ### **Weaknesses 1**
>
> > Please correct me if I have overlooked anything, but an important ablation study seems to be missing. Specifically, an ablation on the two newly introduced terms in Equation (7)—the policy regularization term and the value-based guidance term—would be helpful. What would be the impact on performance if either of these two terms were removed?
>
> **Response:**
> We are happy to provide clarification on this point. In fact, the ablation study you mentioned is included as part of our analysis in Section 5.3. The two newly introduced terms, (1) the policy regularization term and (2) the value-based guidance term, correspond to the Explicit Rebal. (Explicit Rebalancing Mechanism) and Implicit Rebal. (Implicit Rebalancing Mechanism) presented in Table 5, respectively. In Table 5, the variant “RDT” represents the ablated version of RVDT without the value-based guidance component, while “VDT” corresponds to the version without the policy regularization term.
>
> Table 5: Component-level breakdown across ablation variants
> |Component|DT|DT-Dup|**RDT**|QT|**VDT**|RVDT-Dup|RVDT-Determ|RVDT|
> |-|-|-|-|-|-|-|-|-|
> |Explicit Rebal.|None|Dup.|**KL**|None|**None**|Dup.|KL|KL|
> |Implicit Rebal.|None|None|**None**|Q-value|**Q-value**|Q-value|Q-value|Q-value|
> |Policy Type|Determ.|Stoch.|**Stoch.**|Determ.|**Stoch.**|Stoch.|Determ.|Stoch.|
>
> The impact of removing either component can be observed in Table 6 and is discussed in Lines 344–347. Specifically, as stated: “Both RDT (explicit rebalancing) and QT/VDT (implicit rebalancing) individually outperform DT, confirming the effectiveness of applying return-coverage rebalancing mechanisms on DT, and indicating that the combination of rebalancing mechanisms in RVDT provides additional performance gains.”
>
> Table 6: Performance comparison of ablation variants across MuJoCo tasks
> |Task|DT|DT-Dup|**RDT**|QT|**VDT**|RVDT-Dup|RVDT-Determ|RVDT|
> |-|-|-|-|-|-|-|-|-|
> |halfcheetah|84.2|90.3|**90.5**|91.2|**89.5**|93.4|91.5|94.9|
> |hopper|109.5|112.1|**111.9**|112.3|**112.6**|112.1|113.6|113.8|
> |walker2d|108.2|108.8|**109.7**|113.2|**110.3**|110.9|113.1|118.7|
> |Average|100.6|103.7|**104.0**|105.6|**104.1**|105.5|106.1|109.1|
>
> We will revise the manuscript to make this correspondence clearer.
>
> ---
>
> ### **Weaknesses 2: RVDT vs. D-QL in Larger Maze-like Environments**
>
> > In the experimental section, the authors attribute RVDT's inferior performance compared to D-QL on the larger AntMaze tasks to D-QL's highly expressive policies. However, due to the lack of further experimental evidence, it would be helpful if the authors could clarify how D-QL performs relative to RVDT on larger Maze2D tasks, which are similarly large maze-like environments. Alternatively, could the authors provide additional qualitative or quantitative results to support this claim?
>
> **Response:**
> We sincerely thank the reviewer for the thoughtful observation and helpful suggestion.
> Actually, compared to RVDT, D-QL leverages a more expressive diffusion policy, but lacks the long-term modeling capability offered by conditional sequence modeling and the return-coverage rebalancing mechanism we propose.
> To provide additional clarity, we present the performance of D-QL on larger Maze2D tasks below:
>
> Table: Performance comparison on Maze2D Tasks.
> |Maze2D Tasks|CQL|IQL|D-QL|RVDT|
> |-|-|-|-|-|
> |maze2d-m|41.8|34.9|91.0|183.5|
> |maze2d-l|49.6|61.7|200.7|254.3|
>
> From the results above, it can be clearly observed that the D-QL algorithm, which utilizes a diffusion policy, significantly outperforms other Q-learning-based algorithms with simpler Gaussian policies, i.e., CQL and IQL. This highlights the superiority of diffusion policies over Gaussian policies.
>
> Note that, unlike in the AntMaze task, RVDT outperforms D-QL on the Maze2D tasks. This discrepancy may be attributed to the less complex agent used in Maze2D, which is a simpler particle with a relatively small action space. In contrast, the multi-legged agent in AntMaze places greater demands on multi-modality action modeling capabilities. As a result, in the Maze2D tasks, the performance gains from the diffusion policy are less pronounced than those provided by long-term modeling and the return-coverage rebalancing mechanism.
>
> Based on these insights, we conclude that both highly expressive policy architectures and our proposed return-coverage rebalancing mechanism contribute positively to overall algorithm performance. Particularly, in tasks characterized by multi-modal action spaces (e.g., AntMaze), the expressiveness of the policy becomes especially crucial. This conclusion aligns well with the prevailing research direction that emphasizes policy expressiveness. We leave further exploration of combining diffusion policies with return-coverage rebalancing in CSM frameworks as our future work.
>
> ---
>
> ### **Weaknesses 3: Computational Overhead of RVDT**
>
> > What is the computational overhead introduced by RVDT? For instance, RVDT requires first training an expert policy $\pi_e$ and then introduces two additional terms during offline RL training.
>
> **Response:**
>
> We sincerely thank the reviewer for this careful observation. Below, we provide a table detailing the training time comparison of DT, QT, RVDT (including the expert policy $\pi_e$ training phase), and the expert policy training phase itself, measured until the algorithm has converged (i.e., 120 training epochs) on an NVIDIA GeForce RTX 4090 GPU.
>
> Table: Training time comparison, where RVDT's training time includes the expert policy training phase.
> |Gym Tasks|DT|QT|RVDT (total)|Expert policy|
> |-|-|-|-|-|
> |halfcheetah-m-e|7.3hr|12.2hr|15.5hr|3.3hr|
> |hopper-m-e|3.6hr|10.3hr|14.6hr|3.0hr|
> |walker2d-m-e|6.1hr|11.7hr|15.4hr|3.3hr|
> |halfcheetah-m|6.3hr|11.8hr|16.1hr|3.2hr|
> |hopper-m|3.5hr|9.2hr|14.0hr|3.2hr|
> |walker2d-m|6.1hr|11.5hr|16.0hr|3.2hr|
> |halfcheetah-m-r|7.2hr|12.2hr|17.4hr|3.3hr|
> |hopper-m-r|3.5hr|8.5hr|12.3hr|3.2hr|
> |walker2d-m-r|6.2hr|11.2hr|14.8hr|2.8hr|
> |**Average**|**5.5hr**|**11.0hr**|**15.2hr**|**3.2hr**|
>
> As illustrated in the table, the expert policy training phase typically accounts for roughly one-fifth of the total RVDT training time. Excluding this phase, the remaining training overhead introduced by RVDT relative to QT is less than 10%. In addition, we observe that QT already introduces a significant increase in training time compared to DT.
>
> It is important to note that the primary objective of our work is to derive high-performing policies from fixed datasets. As a result, the extended training time incurred by RVDT represents a modest trade-off, especially given the significant performance gains it delivers over other CSM-based methods. The resulting improvements in policy quality outweigh the additional computational cost.
>
> ---
>
> ### **Weaknesses 4: Related Work on Expert Data Usage**
>
> > Some related works also utilize a small portion of expert data to improve offline RL performance [1,2,3].
>
> **Response:**
>
> We thank the reviewer for bringing these relevant references to our attention. These works expand the scope of expert data usage in offline RL across different settings.
>
> Specifically, **Kumar et al.** propose PTR [1], a meta-learning-inspired framework that first performs multi-task offline RL pretraining on diverse robotic demonstration datasets, then fine-tunes on target tasks using only 10-15 expert demonstrations. **Yang et al.** propose GORL [2], a plug-in framework that adaptively balances policy improvement and constraint for each training sample by employing a guiding network trained with only a small number of expert demonstrations. The guiding network is meta-optimized in a MAML-like manner, alternating between updates on expert data and policy learning on the offline dataset. **Cheng et al.** propose TELS [3], which aims to enable effective policy learning in low-data regimes (e.g., 1% of the original dataset). In summary, these settings include pretraining and fine-tuning approaches for robotics [1], meta-learning approaches [1,2], and sample-efficient offline RL with limited data [3].
>
> Our method focuses on leveraging expert data in a more general offline RL setting, which differs from the settings for these works. We acknowledge that these works also explore the utilization of expert data under more complex or specialized scenarios. We will add an additional discussion of methods that utilize a small portion of expert data under broader or task-specific problem formulations in our revision.
>
> [1] Kumar, Aviral, et al. "Pre-training for robots: Offline RL enables learning new tasks from a handful of trials." arXiv preprint arXiv:2210.05178 (2022).
>
> [2] Yang, Qisen, et al. "Hundreds guide millions: Adaptive offline reinforcement learning with expert guidance." IEEE Transactions on Neural Networks and Learning Systems 35.11 (2023): 16288–16300.
>
> [3] Cheng, Peng, et al. "Pushing the Limit of Sample-Efficient Offline Reinforcement Learning." ICLR 2025 Workshop on World Models: Understanding, Modelling and Scaling.
>
> ---
>
> Again, we greatly appreciate the reviewer’s helpful suggestions and the opportunity to clarify and improve our work. We hope our response can alleviate some of your concerns and offer useful clarification.

---

> > ### Comment · Reviewer_Lvw8 · 2025-08-06
> >
> > Thank you for your detailed responses, which have addressed most of my concerns. However, the additional computational overhead appears to offer limited cost-effectiveness. I will keep my original rating but increase my confidence score to reflect a leaning toward acceptance.

---

> > > ### Author Response · Authors · 2025-08-07
> > >
> > > We thank the reviewer for taking the time to review our work. We are glad that our clarifications have effectively addressed your concerns.
> > >
> > > Regarding the reviewer’s remaining point on the computational overhead and its associated benefits, we fully acknowledge that RVDT shows modest improvements over QT on certain tasks (e.g., MuJoCo). This phenomenon mainly results from boundary effects, as both methods already achieve or surpass the expert-level returns provided by the dataset, leaving limited room for further improvement.
> > >
> > > Nonetheless, the advantage of RVDT over QT becomes especially evident in scenarios where accurate Q-value estimation is challenging, such as low-data or sparse-reward settings. As demonstrated in Table 3 (lines 311–320), RVDT significantly outperforms QT under low-data regimes. Specifically, these four datasets contain approximately one-tenth of the original D4RL data, with task difficulty increasing progressively from $D_1$ to $D_4$ (see Appendix F.1 for details). Under these conditions, RVDT consistently achieves average improvements of 45.4%, 14.5%, 14.2%, and 45.6% over QT on $D_1$–$D_4$, respectively. Further evidence of RVDT’s effectiveness in sparse-reward settings is presented in Table 4 (lines 321–329), showing improvements of 27%, 38%, 7%, and 6% over QT on maze2d-open, maze2d-umaze, maze2d-medium, and maze2d-large, respectively. Collectively, these results clearly demonstrate RVDT’s consistent advantages, particularly on challenging tasks.
> > >
> > > Table 3: Performance comparison in low-data regimes.
> > > |Task|QT($D_1$)|RVDT($D_1$)|QT($D_2$)|RVDT($D_2$)|QT($D_3$)|RVDT($D_3$)|QT($D_4$)|RVDT($D_4$)|
> > > |-|-|-|-|-|-|-|-|-|
> > > |maze2d-u|100|171|81|101|73|77|61|100|
> > > |maze2d-m|137|187|175|190|163|182|98|175|
> > > |maze2d-l|109|140|81|90|104|131|100|95|
> > > |Average|115|166|112|127|113|130|86|124|
> > >
> > > Note that the primary objective of offline RL is to derive policies with the highest possible performance from fixed datasets. Thus, the additional computational cost incurred by RVDT can be viewed as a modest trade-off, especially considering the substantial performance improvements it achieves on challenging tasks.
> > >
> > > Finally, we sincerely appreciate the reviewer’s valuable feedback and your decision to increase the confidence score to reflect a leaning toward acceptance. We remain open and happy to discuss further if you have any additional questions.

---

### Comment · Area_Chair_7gjS · 2025-08-06
**Reviewers, please be reminded that ...**

as announced by program chairs, “Mandatory Acknowledgement” button is to be submitted only when reviewers fulfill all conditions below (conditions in the acknowledgment form):
* read the author rebuttal
* engage in discussions (reviewers must talk to authors, and optionally to other reviewers and AC - ask questions, listen to answers, and respond to authors)
* fill in "Final Justification" text box and update “Rating” accordingly (this can be done upon convergence - reviewer must communicate with authors first)

---

### Note · Authors · 2025-08-12

Dear Reviewers, AC, SAC, and PC,

We sincerely thank you for your constructive engagement, which has improved our paper's clarity and completeness.

We are glad that all reviewers have reached consensus and are willing to provide positive recommendations for acceptance. We would like to first remind you of our key contributions. The primary contribution lies in the theoretical investigation of how the performance of CSM-based policies is influenced by the return-coverage of both expert-level and full-spectrum conditional returns in offline RL. Based on our theoretical analysis, we propose a simple yet effective rebalancing mechanism, which is lightweight and can be seamlessly integrated into existing CSM-based algorithms to enhance robustness and performance. The resulting algorithm (RVDT) achieves SOTA performance on the D4RL benchmark.

In response to the reviewers' comments, we have addressed the following main concerns:

Return-coverage assumption ([8NTa]): The return-coverage assumption in our manuscript is widely adopted and recognized as standard and reasonable within the CSM literature. It reflects the inherent nature of CSM-based methods: when the return coverage of the training data is insufficient, reliable policy learning cannot be guaranteed, and the performance gap can become arbitrarily large. Our clarification was acknowledged by Reviewer [8NTa], who expressed agreement and indicated a willingness to raise the score.

Additional computational cost ([Lvw8] and [8NTa]): The primary goal of our work is to obtain high-performing policies from fixed datasets. The additional training time introduced by RVDT (approximately 3 hours per task) represents a modest trade-off given the performance gains over other CSM-based methods.

Marginal improvement ([8NTa] and [Mnay]): The modest gains of RVDT over QT on certain tasks (e.g., MuJoCo) mainly result from boundary effects: both methods already reach or exceed the dataset's expert-level returns, leaving limited room for further improvement. Nevertheless, RVDT's advantage becomes especially evident in scenarios where accurate Q-value estimation is challenging, such as low-data or sparse-reward settings, with relevant supporting experiments provided in Table 3 and Table 4.

Overall, we have made every effort to address all concerns raised during the rebuttal and have received uniformly positive feedback.

We sincerely thank all reviewers for their time, effort, and constructive feedback.

---

### Decision · Program_Chairs · 2025-09-17

**Decision:**

Accept (poster)

**Comment:**

(a) Summary

The paper analyzes why conditional sequence modeling (CSM) policies in offline RL succeed or fail by linking performance to two kinds of return coverage—expert-level and full-spectrum—and derives bounds and propositions formalizing this dependency. Building on the analysis, it proposes RVDT, a DT-based objective combining expert-policy KL (explicit rebalancing) with Q-guided regularization (implicit rebalancing). Across D4RL (Gym, Kitchen, AntMaze, Maze2D), RVDT matches or exceeds strong CSM baselines and is notably better in low-data and sparse-reward settings.

(b) Strengths
- The paper clearly identifies and studies the impact of expert-level and full-spectrum return coverage on CSM performance, providing useful theory. [Lvw8][4BE1]
- The RVDT objective is simple to implement yet effective, yielding strong results and clean ablations. [Lvw8][Mnay]
- The experiments are broad and systematic, covering data quality, scale (low-data), and sparse-reward regimes. [Lvw8][4BE1]
- Writing and presentation are clear, making complex material accessible. [Mnay]

(c) Weaknesses
- Theoretical guarantees hinge on a return-coverage assumption that can be weak or vacuous when coverage is tiny. [8NTa]
- Gains over QT are modest on several Gym/Kitchen tasks, and the method is close in spirit to QT. [Mnay][8NTa]
- The method depends on selecting “expert” trajectories, which may be ambiguous in some real-world domains, and KL regularization does not expand support into unseen state-action regions. [4BE1]
- Extra compute is required (training an expert policy and two regularizers), raising cost-effectiveness questions. [Lvw8][8NTa]

(d) Discussion summary

Most of the questions and comments were satisfactorily addressed in the author rebuttal

(e) Decision rationale

The work offers (1) a clear theoretical account of return-coverage in CSM and a practical, plug-in rebalancing mechanism; (2) consistent improvements with especially strong margins in low-data and sparse-reward settings; and (3) thorough ablations and added analyses that addressed reviewers’ core questions.